# Heisenberg-scaling measurement of the single-photon Kerr non-linearity using mixed states

Geng Chen[1,2], Nati Aharon[3], Yong-Nan Sun[1,2], Zi-Huai Zhang[1,2], Wen-Hao Zhang[1,2], De-Yong He[1,2], Jian-Shun Tang[1,2], Xiao-Ye Xu[1,2], Yaron Kedem[4], Chuan-Feng Li[1,2] & Guang-Can Guo[1,2]

Improving the precision of measurements is a significant scientific challenge. Previous works suggest that in a photon-coupling scenario the quantum fisher information shows a quantum-enhanced scaling of $N^2$, which in theory allows a better-than-classical scaling in practical measurements. In this work, utilizing mixed states with a large uncertainty and a post-selection of an additional pure system, we present a scheme to extract this amount of quantum fisher information and experimentally attain a practical Heisenberg scaling. We performed a measurement of a single-photon's Kerr non-linearity with a Heisenberg scaling, where an ultra-small Kerr phase of $\simeq 6 \times 10^{-8}$ rad was observed with a precision of $\simeq 3.6 \times 10^{-10}$ rad. From the use of mixed states, the upper bound of quantum fisher information is improved to $2N^2$. Moreover, by using an imaginary weak-value the scheme is robust to noise originating from the self-phase modulation.

[1] CAS Key Laboratory of Quantum Information, University of Science and Technology of China, Hefei 230026, China. [2] Synergetic Innovation Center of Quantum Information and Quantum Physics, University of Science and Technology of China, Hefei 230026, China. [3] Racah Institute of Physics, The Hebrew University of Jerusalem, Jerusalem 91904, Givat Ram, Israel. [4] Department of Physics, AlbaNova University Center, Stockholm University, 106 91 Stockholm Sweden. Correspondence and requests for materials should be addressed to Y.K. (email: yaron.kedem@fysik.su.se) or to C.-F.L. (email: cfli@ustc.edu.cn)

Consider a physical process that is described by an interaction Hamiltonian $gH$, which depends linearly on a small parameter $g$ that we want to estimate. The precision of this estimation is ultimately limited by the Cramér-Rao bound, which implies that[1]

$$\Delta g \geq \Delta g_{\min} = \frac{1}{\sqrt{F(\rho)\tilde{\nu}}}, \qquad (1)$$

where $F(\rho)$ is the quantum Fisher information (QFI) of the final state, $\rho$, and $\tilde{\nu}$ is the number of times $H$ is used. For pure states the QFI is equal to $4\Delta H^2$, where $\Delta H = \sqrt{\langle H^2 \rangle - \langle H \rangle^2}$ is the standard deviation of $H$ with respect to the initial state. Hence, by preparing an initial pure state with a large $\Delta H$, one may improve the precision. The interaction can involve a large number, $N$, of subsystems, and in case that there are no interactions between the subsystems, $H = \sum_{i=1}^{N} H_i$. The scaling of the precision with respect to $N$ is of special importance. Previous works show that when some quantum resources are utilized, e.g., quantum entanglement, it is possible to have $\Delta H \propto N$, which yields an Heisenberg scaling (HS)[2–6]. Thus, a significant amount of effort was put into generating highly entangled states, such as NOON states[7–9] or squeezed states[10–13]. Zhang et al. also prove that on the appearance of photon-coupling, the QFI can also shows a quantum scaling of $N^2$, even without any quantum resources[14]. However, as far as we know, no experimental works have been demonstrated to extract this amount of QFI and eventually reach a realistic HS.

Generally, for mixed states Eq. (1) does not hold; an initial mixed state with $\Delta H \propto N$, does not yield an HS. It has been shown, however, that for non-linear interactions the precision can attain HS with mixed states[15]. Our scheme is directly focused on maximizing $\Delta H$ by introducing externally induced fluctuations to the initial probe state. In order to see how to utilize these fluctuations, we use the formalism of weak measurements[16–22]. Consider an interaction Hamiltonian $H = f(t)\hat{P}\hat{C}$, where $\hat{P}$, which is related to a probe, and $\hat{C}$, which is related to a system, are both Hermitian operators and $f(t)$ is a coupling function with a finite support that satisfies $\int f(t)dt = g \ll 1$. If the system is prepared in a state $|\psi\rangle$ before the interaction and post-selected later to a state $|\varphi\rangle$, then $\langle \hat{P} \rangle$ will be modified according to $\langle \hat{P} \rangle \rightarrow \langle \hat{P} \rangle + \delta P$, with

$$\delta P = 2g(\Delta P)^2 \mathrm{Im} C_w, \qquad (2)$$

where $C_w = \frac{\langle \varphi | \hat{C} | \psi \rangle}{\langle \varphi | \psi \rangle}$ is the weak value of $\hat{C}$, and $\Delta P$ is the standard deviation of $\hat{P}$[23,24]. Hence, a measurement of $\langle \hat{P} \rangle$ yields a precision of $\Delta g \propto \Delta P^{-1}$. As we noted before, when $\hat{P}$ pertains to $N$ uncorrelated systems, e.g., a coherent state, then $\Delta P \propto \sqrt{N}$. However, it was shown[25] that Eq. (2) holds even when $\Delta P$ is due to classical fluctuations. By taking the limit of the largest possible fluctuations $\Delta P \sim \langle \hat{P} \rangle$, we can significantly improve the measurement precision.

Remarkably, for the metrological task of estimating a coupling strength between a probe and a pure quantum system, our scheme enables to utilize these classical fluctuations, i.e., mixed states, hence the upper bound of QFI can be raised to $2N^2$ in principle. Altogether with a post-selection process, our method can extract an amount of Fisher information (FI) which is also $\sim N^2$, thus the achieved precision is close to the theoretically optimal precision set by the QFI. We experimentally demonstrate our scheme by measuring a single-photon's Kerr non-linearity[26], achieving a robust HS that results in a precision of $\simeq 3.6 \times 10^{-10}$ rad.

## Results

**Heisenberg-scaling metrology**. The general idea of our scheme is shown in Fig. 1. The intuition arising from this picture, is that 'stretching' the state can, in a particular case, have a similar effect as squeezing[27]. Increasing the total uncertainty is possible when external noise/modulations are added.

Consider the setup shown in Fig. 1a. The interaction is described by $U = e^{-ig\hat{n}\hat{C}}$, where $\hat{n}$ is the photon number operator on the probe and $\hat{C}$ is the 'which path' operator acting on a single photon going through the interferometer. If the photon is in a superposition of the two arms, i.e., not in an eigenstate of $\hat{C}$, the QFI of the joint system-probe state after the interaction is $\propto N^2$, where $N = \langle \hat{n} \rangle$[14]. However, using a probe that is initially in a coherent state, a measurement of any other quadrature, cannot extract this information, since HS arises only in the post-selection process itself[14,25,28]. In this experiment, we measured the photon number, for a given post-selection. When the probe is initially in a statistical ensemble with a wide distribution, i.e., the standard deviation is proportional to $N$, the classic Fisher information[29] for this choice is also $\propto N^2$, regardless of the particular form of the distribution (see Supplementary Note 2).

If the system is prepared in a state $|\psi\rangle$ and post-selected later to $|\varphi\rangle$, one can approximate the impact of the interaction as $\langle \varphi | e^{-ig\hat{n}\hat{C}} | \psi \rangle \simeq e^{-ig\hat{n}C_w}$. A coherent state would transform

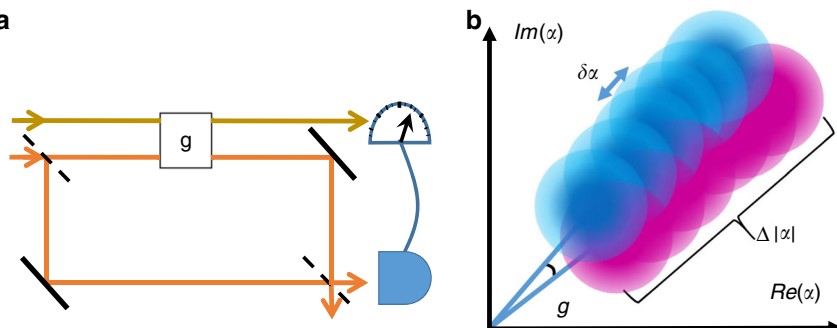

**Fig. 1** Strategies for precision measurements. **a** The scheme: a single-photon goes through an interferometer, where in one arm it interacts with a coherent state and a phase is acquired, while in the other nothing happens. At the exit port the two paths interfere so the probability of the photon coming out there depends on the phase it acquired by the interaction, and thus, also on the number of photons in the probe. Post-selecting the probe pulses accordingly induces a shift in the average photon number, from which the interaction strength can be estimated. **b** The principle of our method: a statistical ensemble of $|\alpha\rangle$ (only a few are drawn), showing the two paths for each member: blue with the phase and magenta without. Due to the probability of postselection, Eq. (3), the ensemble is shifted in the radial direction. The shift in photon number, Eq. (5), $\delta n \propto (\Delta n)^2 \propto N^2$ so the shift in $|\alpha|$ is $\propto N^{3/2}$. The uncertainty in $n$ can be fluctuated to be $\propto N$. The ratio between the uncertainty and the shift is $\propto 1/N$, resulting in the Heisenberg scaling

according to $|\alpha\rangle \rightarrow |e^{igC_w}\alpha\rangle$, which implies a tangential shift of $g|\alpha|\mathrm{Re}C_w$ and a radial shift of $g|\alpha|\mathrm{Im}C_w$, yielding a precision $\propto 1/\sqrt{N}$. In our method, we use a statistical mixture of number states or an ensemble of coherent states, as illustrated in Fig. 1b. The probability of post-selection when the probe has $n$ photons is given by (see Supplementary Note 2)

$$p_{\varphi|n} = |\langle\varphi|\psi\rangle|^2(1 + 2g\mathrm{Im}C_w n) + o(ng)^2. \quad (3)$$

States with a higher photon number entail larger (smaller) probability if $\mathrm{Im}C_w > 0$ ($\mathrm{Im}C_w < 0$); therefore the photon-number distribution is shifted. This change in the average photon number is given by $\delta n = 2g(\Delta n)^2\mathrm{Im}C_w$, where $\Delta n$ is the standard deviation of the initial distribution. Obtaining $\delta n$ experimentally entails a measurement error $\propto \Delta n$, and thus, the estimation $g = \delta n/(2(\Delta n)^2\mathrm{Im}C_w)$ yields a precision $\Delta g \propto 1/\Delta n$ (other contributions to the estimation error are $O(\delta n/\Delta n) \ll 1$). A coherent state $|\alpha\rangle$ has an uncertainty of $\Delta n = |\alpha| = \sqrt{N}$, but adding classical fluctuations one can produce a distribution with a large deviation, $\Delta n \propto N$, where $N$ is the average photon number of the distribution. One can separate the uncertainty to a component coming from the quantum state and another coming from the statistical distribution. In our case the latter is much larger so the former can be neglected. Note that the precise form of the distribution is insignificant for this result; it depends only on the mean value and the variance of the distribution (see Supplementary Note 2).

**Kerr non-linearity measurement**. Let us apply this method in the task of measuring the Kerr non-linearity of a single photon. A strong pulse (probe) and a single photon (system) overlap inside a fiber where the dependence of the refractive index on the intensity of light induces both a self-phase modulation (SPM) and a cross-phase modulation (XPM). The effective Hamiltonian is[30] $H = \tilde{g}_S\hat{n}^2 + \tilde{g}\hat{C}\hat{n}$, where $\hat{n}$ now refers to the photon number in the probe, $\hat{C}$ is the photon number in the system, and $\tilde{g}_S$ ($\tilde{g}$) is the coupling due to the SPM (XPM). Integrating along the fiber yields the coefficient $g_S$ ($g$) for the SPM (XPM), such that the evolution is given by $U = e^{-ig_S\hat{n}^2 - ig\hat{C}\hat{n}}$. The XPM represents an interaction involving a single photon, and thus, measuring $g$ is highly important for many applications[31,32]. An experiment to achieve this was recently performed by Matsuda et al.[26], using the standard approach as described above. The main limitation in their setup came from the additional noise introduced by the SPM, which is dominant when $Ng_S \sim 1$. Using our scheme, as we show below, the SPM is insignificant since the intensity is measured instead of the phase.

We now show, in detail, how to measure $g$ using our method and analyze the resulting precision. The system photon is sent into an interferometer, with one arm containing the fiber, such that its initial state is $|\psi\rangle = (|1\rangle + |0\rangle)/\sqrt{2}$ where $|1\rangle$ and $|0\rangle$ are eigenstates of $\hat{C}$ with eigenvalues 0 and 1, respectively. The probe is in a coherent state $|\alpha\rangle$, but by modulating the power of the laser we obtain a distribution of $\alpha$, which can be written as a mixed state, $\rho_p = \sum_\alpha p_\alpha |\alpha\rangle\langle\alpha|$, where $p_\alpha$ is the probability of having a coherent state $|\alpha\rangle$. After the probe goes through the fiber, we measure its average photon number. However, only the trials when the system photon is found in a specific exit port are taken into account; we post-select the state of the system as $|\varphi\rangle = (|1\rangle - e^{-i\varepsilon}|0\rangle)/\sqrt{2}$, with the post-selection parameter $\varepsilon \ll 1$, which is set by tuning the interferometer. The result of the measurement on the probe is given by

$$\langle\hat{n}\rangle = \frac{\mathrm{Tr}\left[\hat{n}U\rho_p\rho_s^\psi U^\dagger\rho_s^\varphi\right]}{\mathrm{Tr}\left[U\rho_p\rho_s^\psi U^\dagger\rho_s^\varphi\right]}, \quad (4)$$

where $\rho_s^\psi = |\psi\rangle\langle\psi|$ and $\rho_s^\varphi = |\varphi\rangle\langle\varphi|$ are the pre- and post-selected state, respectively. $\rho_s^\varphi$ is inserted to represent the post-selection and this is also the reason for the normalization denominator. Since $[U, \hat{n}] = 0$, only the diagonal element $\rho_p^{n,n} = \langle n|\rho_p|n\rangle$ are significant, and we can replace the trace over the probe states with a sum over the photon number $\mathrm{Tr}_{\rho_p}[\bullet] \rightarrow \sum_n \rho_p^{n,n}[\bullet]$ while replacing $\hat{n} \rightarrow n$ inside the summand. The trace over the system can be approximated as $\mathrm{Tr}\left[U(n)\rho_s^\psi U^\dagger(n)\rho_s^\varphi\right] \simeq \varepsilon^2\left(1 + 2n\frac{g}{\varepsilon}\right)$ for $n\frac{g}{\varepsilon} \ll 1$ (see Supplementary Note 3). Therefore, the change in the average photon number is given by (see Supplementary Note 2)

$$\delta n \simeq \frac{\sum_n \rho_p^{n,n}n\left(1 + 2n\frac{g}{\varepsilon}\right)}{\sum_n \rho_p^{n,n}\left(1 + 2n\frac{g}{\varepsilon}\right)} - N \simeq 2\frac{g}{\varepsilon}(\Delta n)^2. \quad (5)$$

Since $C_w \simeq \frac{i}{\varepsilon}$, Eq. (5) agrees with Eq. (2). In case that $\Delta n \propto N$, we obtain an HS. To describe our scheme in a more general context, we also present a detailed derivation for the case in which the system and probes are qubits (see Supplementary Note 4).

Moreover, due to the usage of an imaginary weak value, the interaction results in a shift of the average photon number rather than a phase shift, our scheme is robust to phase noise, and in particular, the SPM part is completely canceled.

Our method, and weak measurements in general, requires a post-selection, and for a large weak value, the post-selection is rare. This can diminish the precision due to a decrease in the number of successful post-selecting events $\nu \sim \tilde{\nu}|\langle\varphi|\psi\rangle|^2$[33,34]. On the other hand, by calculating the Fisher information directly from Eq. (5), one obtains another amplification factor of $\varepsilon^{-2} \sim |\langle\varphi|\psi\rangle|^{-2}$; therefore, when using the Cramér-Rao bound Eq. (1), the dependence on $|\langle\varphi|\psi\rangle|$ cancels.

The experimental setup is shown in Fig. 2. A photon pair is generated by spontaneous parametric down conversion. One

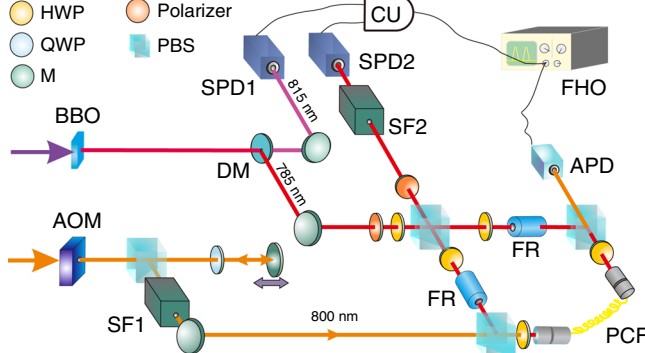

**Fig. 2** Experimental setup for measuring the Kerr non-linearity of a single photon. Photon pairs (with wavelengths 785 and 815 nm) are prepared through the spontaneous parametric down conversion process by pumping the BBO crystal with ultraviolet pulses. The 815 nm photons serve as triggers and the heralded 785 nm photons interact with strong probe pulses (800 nm) in an 8 m long photonic crystal fiber (PCF). The acoustic optical modulator (AOM) is used to introduce fluctuation to the probe via modulation in the intensity. After interaction, the state of the single photon is post-selected and the corresponding strong pulses are measured using a full HD oscilloscope (FHO). Analyzing the shift of average photon number of these strong pulses yields an estimation for the interaction strength. For more details, see the main text and Method. BBO -$\beta$-barium borate crystal, DM dichroic mirror, SPD: single-photon detector, HWP: half wave plate, QWP: quarter wave plate, M: mirror, FR: Faraday rotator, SF: spectral filter, PBS: polarized beam splitter, APD: amplified photon detector, CU: coincidence unit

photon is used for heralding and the other enters a polarization Sagnac interferometer (PSI), which contains a photonic crystal fiber (PCF)[35]. The single-photon's polarization is set as $(|V\rangle + |H\rangle)/\sqrt{2}$ ($V$ and $H$ represent the vertical and horizontal polarization, respectively). After entering the PSI, the photon is in an equal superposition of clockwise and counter-clockwise propagation. Only the counter-clockwise component can interact with the probe pulse; hence, the system becomes $|\psi\rangle = (|1\rangle_V + |0\rangle_H)/\sqrt{2}$, where {0, 1} represents the interacting photon number. After the PSI the system is post-selected using its polarization. Faraday units cause the two components to have the same polarization inside the PCF. Preparation of the probe, which is a strong pulse, involves (i) modulating its intensity using an acoustic optical modulator (AOM), (ii) introducing delay using a translatable mirror and (iii) filtering the spectrum to prevent an overlap with the spectrum of the single photon. The probe then enters the PCF through a polarized beam splitter (PBS), where it overlaps with one component of the single photon and the interaction takes place. Upon exiting the PCF, through another PBS, the intensity of the probe is measured, which depends on the detection of both the heralding photon and the post-selected photon from the PSI. Separating the single photon from the strong pulse after the interaction is performed using both the polarization and spectrum degrees of freedom (see the Methods for more details).

**Experimental results.** We start by demonstrating the validity of Eq. (5) in our system by modifying the quantities on the right side: the interaction strength $g$, the standard deviation $\Delta n$ and the weak value $i/\varepsilon$, and by measuring $\delta n$. Tuning $g$ is performed by varying the temporal overlap between the probe and the single photon. The standard deviation is controlled by changing the modulation amplitude $D$ in the intensity of the probe (see the Methods for details). $\varepsilon$ is set by choosing the post-selected polarization state of the single photon exiting the PSI. In Fig. 3, we plot the normalized change in the photon number $\delta\tilde{n} = \delta n/N$ in a number of ways. The error bars are shown as the uncertainty in $\delta\tilde{n}$, which is written as $\sigma/\sqrt{\nu}$. Here, $\sigma$ is the standard deviation of measured $\delta\tilde{n}$ and $\nu$ is number of recorded probe pulses by FHO. The results demonstrate the ability to detect the interaction of the probe with a single photon.

We now experimentally demonstrate the precision of our method, and, in particular, we show how the precision scales with the average photon number of the the probe. Theoretically, the precision can be obtained from Eq. (5), for example, by

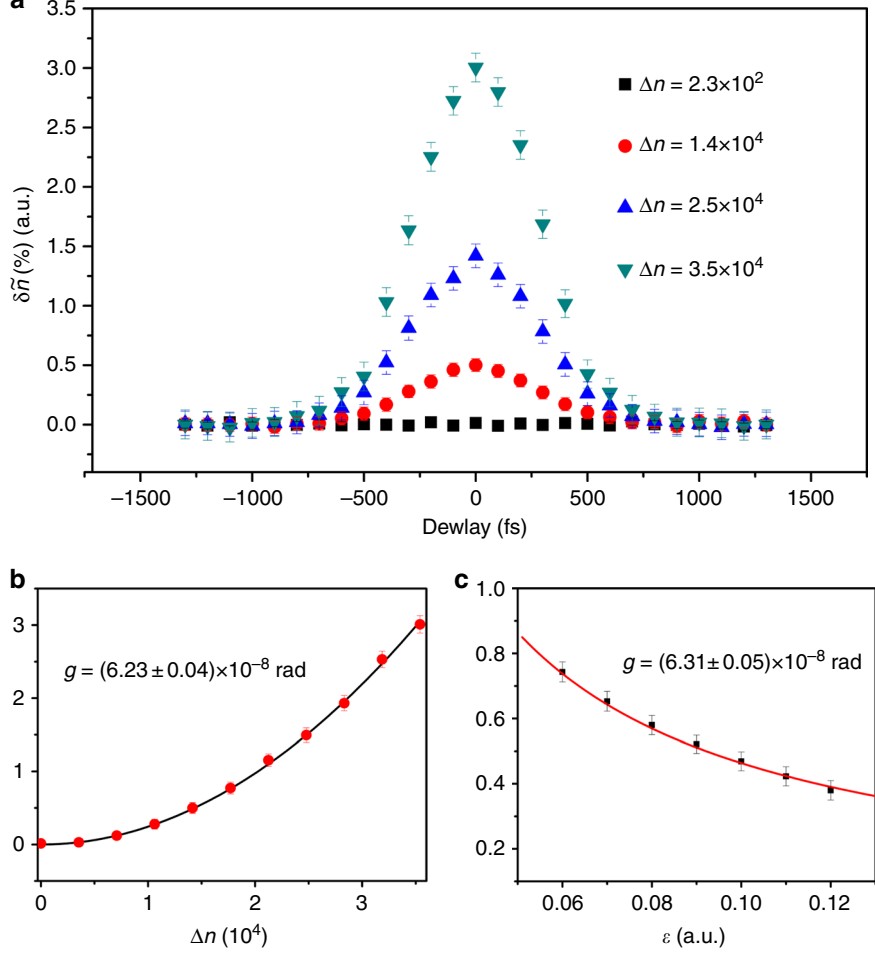

**Fig. 3** Demonstration of the validity of Eq. (5). All plots show the normalized change in the average photon number $\delta\tilde{n}$ with $N = 5 \times 10^4$. **a** A delay in the probe controls the temporal overlap with the system inside the PCF and thus tunes the interaction strength (here with $\varepsilon = 0.1$). This is performed for several magnitudes of modulation, and the results trace out Gaussian shape for non-zero modulation amplitude. However, in the absence of modulation, the standard deviation due to the shot noise $\Delta n = \sqrt{N} \simeq 10^2$ is too small to observe the effect of the interaction. **b** The standard deviation $\Delta n$ is changed by controlling the magnitude of the modulation in the intensity of the probe, with $\varepsilon = 0.1$. **c** Tuning the post-selection parameter $\varepsilon$, while $\Delta n = 1.4 \times 10^4$. The results in both (**b**, and **c**) are fitted to Eq. 5, and an estimate of $g$, with an error due to the fitting quality, is shown in each panel. The error bars are shown as the uncertainty in $\delta\tilde{n}$, which is written as $\sigma/\sqrt{\nu}$. Here, $\sigma$ is the standard deviation of measured $\delta\tilde{n}$ and $\nu$ is number of recorded probe pulses by FHO

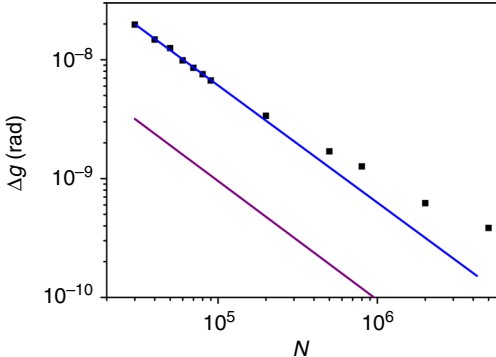

**Fig. 4** Experimentally obtained precision showing Heisenberg scaling. By varying the interaction parameter $g$, via tuning of the temporal overlap of the system and probe, we obtained $s = \frac{\partial \delta \bar{n}}{\partial g}$ (see Supplementary Note 5 for details). The precision of the estimation procedure is then given by $\Delta g = 2\sigma/s\sqrt{\nu}$. We performed this using $\varepsilon = 0.1$, $\Delta n \simeq 0.5N$, $\nu \simeq 2.2 \times 10^5$ and for a number of values of $N$ from $3 \times 10^4$ to $5 \times 10^6$. For $N < 10^5$ the precision follows an HS, of $\Delta g \simeq 6.3 \times 10^{-4} N^{-1}$ rad, shown as a blue line, obtained by fitting these points. In this regime $s$ has a linear dependency on $N$, while the uncertainty $\sigma/\sqrt{\nu}$ is roughly independent of $N$. For higher values of $N$, there is a deviation from the HS, due to the non-linearity of $s$. The purple line is a bound on the precision for mixed states, taking account the QFI for each member in the ensemble, given by $\Delta g_{\min} \simeq 0.95 \times 10^{-4} N^{-1}$ rad (See Supplementary Note 1 for details)

calculating the Fisher information (see Supplementary Notes 1 and 2). Nonetheless, it is important to show that the precision can be reached in practice, for large values of the photon number, and that the method is indeed advantageous compared to the alternative methods. In order to quantify the precision, we study the dependence of the measured quantity on the varying estimated parameter $g$. Each value of $g$ is calibrated from Eq. (5) with $N = 9 \times 10^4$, $\varepsilon = 0.1$ and $\Delta n \simeq 0.5N$, when $\nu \simeq 9 \times 10^5$. Taking into account of the uncertainty in the measurement of the probe, we can obtain the practical precision. The results, shown in Fig. 4, demonstrate an HS, up to values of $N \approx 10^5$. The ultimate precision of $\Delta g \simeq 3.6 \times 10^{-10}$ rad is an improvement on a recent result for the same task[26]. Considering a pure probe of coherent state gives a QFI $\propto N^2$[14], the bound to QFI with mixed state can be calculated from Supplementary Eq. (3) in Supplementary Note 1. It can be seen that the achieved precisions in our experiment are close to the optimal values (the purple line) set by this bound. From a theoretical point of view, increasing the variance up to $\propto N^2$ can give a bound that is $\propto 2N^2$, which improves the upper bound to the QFI.

## Discussion

The method we presented requires $Ng \ll 1$, which implies that it cannot be a single-shot measurement. The information regarding $g$ can only be gathered from an ensemble that is large enough for the statistical distribution of the initial probe state to be meaningful. Nonetheless, at the most interesting scenario $g \to 0$, i.e., detecting the utmost miniscule effects, the scaling implies that when $g \to \tilde{g} = a^{-1}g$, modifying $N \to \tilde{N} = aN$ would maintain the same relative precision, for any number $a$.

In summary, in this work a new scheme for the metrological task of estimating a weak coupling strength between a pure quantum system and a probe was presented. We theoretically and experimentally demonstrated that mixed probe state, combined with a post-selection of the pure quantum system, can be utilized to improve the precision. Specifically, the extracted FI is close to and scales as the QFI. We performed a measurement of the Kerr non-linearity of a single photon with an HS. Moreover, because

an imaginary weak value was employed, our measurement was robust to phase noise, and in particular, to SPM noise. This enabled us to reach a precision of $\simeq 3.6 \times 10^{-10}$ rad in a measurement of an ultra-small Kerr phase of $\simeq 6 \times 10^{-8}$ rad. Enhancing the precision with weak measurement has been theoretically investigated in the context of metrology[28,36–38], and some other works questioned this advantage considering the discarded resources[14,33]. In our scheme, the mixed states increase the variance of the Hamiltonian, and weak measurements enable the increased variance to improve the precision. The classical noise is not crucial from a theoretical point of view but is vital for the practical method we used. This new technique further develops the theoretical framework of weak measurement. The maximal possible magnitude of fluctuations is limited by the experimental resources, which, in our case, is the photon number. The precision scales inversely with the magnitude of fluctuations. Thus, when the resource is scaled up, the precision improves towards the QFI limit; in our case reaching an HS. The fact that our scheme is based on the utilization of mixed states enables its practical scalability (up to the limits of the scheme). Hence, our work paves a new route for precision measurements, which can significantly modify the vast amount of effort devoted to this task.

## Methods

**Preparation of system and probe.** Single photons (systems) are generated by a non-degenerate spontaneous parametric down conversion process (SPDC). At first, 130 fs laser pulses centered at approximately 800 nm are up-converted to 400 nm by a second harmonic generation (SHG) process in a $\beta$-barium borate (BBO) crystal. Afterwards, a second BBO crystal is pumped by the 400 nm pulses to generate down-converted photon pairs. The cut angle of BBO crystal is designed to generate collinear 785 and 815 nm photon pairs. The photons propagate collinearly and are then separated by a dichroic mirror. The 785 nm photon is reflected into the interferometer as a system pulse while the 815 nm photon is transmitted and then detected by the first single-photon detector to herald the 785 nm photons. The residual 800 nm laser pulse after the SHG process is attenuated as a probe pulse. A feedback control of the probe pulse is realized by a half wave plate (HWP) mounted in a motored rotation stage, so that the power of the probe pulse is well stabilized. Before the probe pulse is coupled into the PCF, its amplitude, spectrum and time domain are tuned. The amplitude modulation is performed through an AOM placed at the confocal point of a doublet lens. The AOM is driven by an arbitrary wave generator (Tektronics AWG 3252). The driving frequency is 200 MHz to maximize the diffraction efficiency. The specific form of the driving function is not important and only the standard deviation affects. For convenience we apply a sine-type modulation on the driver described as $V = V_0(1 - D\sin(\omega t))$, the photon number fluctuates around the mean $N$ with a standard deviation decided by $D$. $V_0$ is selected where the diffraction efficiency is approximately half of the maximum and $N$ is determined by the incident power on AOM. The modulation frequency $\omega$ is fixed at 1 KHz. The modulation depth D can be varied from 0 to 1, as a result, $\Delta n$ can be tuned to expected values according to the records from FHO. The first diffraction order is isolated by a pin-hole and delayed to overlap with pump photons inside the PCF. This synchronization is realized by a silver mirror on a manual linear translation stage with the precision of ~ 10 femtosecond. The first spectrum filter (SF1) is an optical 4-$f$ system including two transmitting gratings (1200 Grooves/mm) and a pair of lenses (300 mm focus length). By aligning a silt on the confocal plane, short wavelengths below 795 nm are filtered. Consequently the system and probe pulses can be separated in the spectrum, which is essential when implementing post-selection on system photons.

**Interaction in PSI.** The initial state of the system photons is prepared as $|\psi\rangle = (|V\rangle + |H\rangle)/\sqrt{2}$, where $H$ and $V$ represent the horizontally and vertically polarized components, respectively. These two components counter-propagate through the PSI, which contains an 8 m long PCF (NL-2.4-800, Blaze Photonics). The incident light is collected into the PCF by two triplet fiber optic collimators (Thorlabs TC12FC-780) lenses with a coupling efficiency of 20%. With a HWP before each collimators, photon polarization is maintained after the PCF. Two Faraday units, each consisting of a 45° Faraday rotator and a HWP, cause the two components to have the same linear polarization in the PCF. Vertically polarized system photons are synchronized to overlap with the probe pulses. Two internal polarized beam splitters are used to allow the system photons to enter and exit the PCF.

**Detection apparatus.** After the probe pulses are separated from the system photons, they shine on a low-noise amplified photon detector and the waveform is sampled by a 12-bit (4096 level) full HD oscilloscope working in the external

trigger mode. The system photon exits the PSI from another port and is then post-selected by a polarizer. The post-selection state $|\varphi\rangle = (|V\rangle - e^{-i\varepsilon}|H\rangle)/\sqrt{2}$ is set to be nearly orthogonal to $|\psi\rangle$. The second spectrum (SF2) filter contains a 4-$f$ system similar to the first one, but here, only photons with wavelengths below 790 nm can pass. As a result, probe photons leaking out of the PSI are filtered. A subsequent 10 nm band pass filter centering at 785 nm and a short-wavelength pass filter cutoff at 790 nm reinforce this filtering. Coincidence signals are used to trigger the FHO and post-select the probe pulses. The final recording rate of post-selected probe pulses is mainly determined by the value of $\varepsilon$. When $\varepsilon$ equals 0.1, data were recorded for a total of 6 h, and ~ 220 K probe pulses waveforms are recorded. The value of $n$ is given by the root mean square (RMS) of the recorded waveform. The uncertainty $\sigma/\sqrt{\nu}$ is also estimated as the standard error of these RMS values.

**Data availability**. The authors declare that all data supporting the findings of this study are available within the article and its Supplementary Information files or from the corresponding author upon reasonable request.

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

## Acknowledgements

This work was supported by the National Key Research and Development Program of China (No. 2017YFA0304100), the National Natural Science Foundation of China (Nos. 61327901, 61490711, 11654002, 11325419, 61308010, 91536219, 11774335), Key Research Program of Frontier Sciences, CAS (No. QYZDY-SSW-SLH003), the Fundamental Research Funds for the Central Universities (No. WK2470000026). N.A. acknowledges the support of the Israel Science Foundation (Grant No. 039-8823) and EU Project DIADEMS.

## Author contributions

Y.K. and N.A. proposed the framework of the theory and made the calculations. C.-F.L. and G.C. planned and designed the experiment. G.C. carried out the experiment assisted by Y.-N.S., X.-Y.X., Z.-H.Z., and W.-H.Z., whereas J.-S.T. designed the computer programs and D.-Y.H. assisted on operating the AWG and FHO. G.C., Y.K., and N.A. analyzed the experimental results and wrote the manuscript. G.-C.G. and C.-F.L. supervised the project. All authors discussed the experimental procedures and results.

## Additional information

**Competing interests:** The authors declare no competing financial interests.

