## [Peer Review File · Nature Communications]

Reviewers' comments:

Reviewer #1 (Remarks to the Author):

Review of NCOMMS-17-08474

Dear editor,

I have read thoroughly the manuscript entitled “Scalable Heisenberg-limited metrology using mixed states”, by Prof Li and co-workers. The paper describes an experiment in quantum metrology using post-selection, where the cross-Kerr nonlinearity between an intense probe field and a single photon is measured. The authors measure the shift in the mean photon number of the probe state, conditioned on a post-selection of the single-photon state. The authors demonstrate that the precision of the scheme scales as $1/N$, where N is the average photon number of the probe field. This is known as Heisenberg-like scaling. This result is rather surprising, since the experiment is performed using a probe that is a mixture of coherent states. A coherent state by itself would give a precision that scales as $1/\sqrt{N}$, known as the standard quantum limit (SQL). For this reason I believe that most physicists in the field would naturally assume that a mixture of coherent states would also follow the SQL. Thus, I found the overall result to be quite interesting.

The result seems to contradict the following intuition: The Fisher information is convex, so the Fisher information available using the ensemble of coherent states is less than or equal to the weighted sum of the Fisher information of each coherent state. Since each coherent state obeys the SQL, how is the Heisenberg scaling reached? Moreover, the authors argue that the classical Fisher information of the scheme is close to the quantum Fisher information by less than an order of magnitude. The theory provided by the authors appears to be correct, so what is wrong with the above reasoning?

I admit that it took some time for me to believe that this result is indeed possible. I would say that the single photon plays an extremely important role here. For example, purity (coherence) of the system photon is required for the scheme to work. It is also necessary that the parameter to be estimated (g) appear in an interaction Hamiltonian between the system and probe, which allows one to take advantage of the coherence of the system. In this regard, I believe that the authors do not emphasize the role of the system photon sufficiently. For example, the title and abstract only mention the mixed probe states. The abstract should mention that the “post-selection” is indeed “post-selection of an additional pure system”, or something along these lines. Surely adding technical noise to a standard experiment would be detrimental. However, with this interaction

Hamiltonian and an additional pure system photon, the classical noise becomes an asset. What additional resource is provided by the system photon?

The experimental is very nice, as are the experimental results. However, the authors should provide a little more discussion in the text concerning their main results in Fig. 4. Do the linear fits in (a) have slope proportional to N ? What is the $1/N$ curve fit in Fig. (b)? How does this compare to the precision limit given by the quantum Fisher information?

There are several conditions for the scheme to work: $\epsilon \ll 1$; $Ng \ll 1$; $ng/\epsilon \ll 1$. It would be interesting if the authors would briefly discuss the theoretical limit to the average photon number that could be used, in the interest of optimizing precision. Do the authors perform the experiment near this value?

In summary, I find the result to be very interesting, but I believe that many readers will struggle to understand why this result is possible. The theory provided in the manuscript is more or less straightforward, but the argument based on simply increasing the variance using classical noise seems to describe something that is too good to be true. Could the authors give a more intuitive argument as to why their scheme achieves Heisenberg scaling? There must be some quantum resource (entanglement? discord? coherence? squeezing?) responsible for surpassing the SQL. What is it?

I believe that the manuscript describes an important and counter-intuitive result that could appear in Nat. Comm, provided the authors can address my comments above. The paper is well-written, as is the supplementary material. Below I mention some additional minor corrections that should be made.

Minor corrections:

Page 2: Why is the probe operator \hat{n} written in operator notation, and the operator C is not?

Page 2, second column: I suggest changing "...we measure it's photon number." to "...we measure it's average photon number".

Methods section: There are a few typos: "secondary harmonic generation" -> "second harmonic generation"; "cutting angle" -> "cut angle", "duplet lens" -> "doublet lens".

I also believe that "the 785 photon is reflection...as a pump pulse" -> "the 785 photon is reflection...as a system pulse"

Detection apparatus section: The post-selection state should be " φ " and not " ψ "

Reviewer #2 (Remarks to the Author):

REVIEW OF NCOMMS-17-08474

1. MAJOR CONCERNS

This paper deals with statistical estimation of g –the strength of a cross-phase modulation in an optical experiment – claiming scalable Heisenberg-limited measurement precision using only mixed states.

If the theory and experiment for this paper stand up to scrutiny, it would seem to be a very significant result, since usually it is believed that a resource such as entanglement is necessary for such a quantum advantage (where the precision scales inverse-linearly rather than inverse-root with the number of photons N). Furthermore, mixed states are generated with classical randomness which is widely believed to reduce quantumness and its associated advantages (whereas here it appears to improve things). Also, the technique presented here is based in the formalism of weak measurement. This suffers from lots of controversy, mainly because of the necessary use of postselection (discarding of data or experimental runs). Postselection is known to be a process that can obscure the true physics and easily lead to nonsensical or misjudged reasoning. It seems then that there is much to be skeptical of in this paper; but seeing as there is also much at stake, let us take a systematic and fair approach to assessing it.

In terms of the quality of writing and presentation: these are on the whole very good with clear figures and a good quality of English. On the other hand, I have some queries and suggestions about specific technical points, and a couple of rather serious objections.

1.1. The main argument. This work is based on theoretical results published in [23], and can be thought of as a protocol which makes two choices: i) to use pre and postselection to arrange a shift in n (by a quantity proportional to $g\Delta n$, where Δn is the uncertainty in photon number) rather than a rotation of the phase (by a quantity proportional to g) and ii) use of classical noise to increase Δn . With all this in mind, in assessing this paper I believe there are at least the following 4 key questions:

1) Does the proposed technique really offer Heisenberg scaling when all resources are taken into account? 2) Does it continue to do so when N is taken arbitrarily high? In other words is it really a scalable technique? 3) Is the classical noise playing a crucial role? 4) Are there theoretically superior techniques that could have been just as easily used?

In response to question 1: I have checked the theoretical proofs from Ref [23] and I find that they are correct in a certain parameter regime. Therefore we can say that there is indeed a Heisenberg scaling with this technique. However, the scaling will not continue forever and therefore we must crucially conclude question 2 in the negative. This is because, as the authors state, their method requires $Ng \ll 1$. for any fixed value of the unknown

Date: May 11, 2017.

parameter g , there is a maximum N such that this condition is satisfied. See Figure 1 of Ref [23], where the signal-to-noise ratio (roughly equivalent to Fisher information) actually starts to decrease when the technical noise (equivalent here to Δn) is increased beyond a certain value. The authors will argue (correctly) that the temporary scaling can survive for longer and longer (for higher and higher values of Δn) as long as g gets smaller and smaller. But since in any real experiment g is finite it is clear that the scaling will always be temporary.

To answer questions 3 and 4, we will proceed by considering an alternative measurement strategy. Consider that the postselected state is but one outcome of a set of operators A_i resolving the identity $\sum_i A_i = \mathbb{I}$. For example, $A_1 = |\phi\rangle\langle\phi|$. The Fisher information associated with the entire (unconditioned) resultant probability distribution will always be greater than or equal to the Fisher information conditioned of one particular outcome, when it has been corrected for the probability of obtaining that outcome:

$$|\langle\phi|\psi\rangle|^2 \nu FI \leq \nu FI \left[\text{Tr} \left((A_i \otimes \hat{n}) U(|\psi\rangle\langle\psi| \otimes \sum_{\alpha} p_{\alpha} |\alpha\rangle\langle\alpha|) U^{\dagger} \right) \right].$$

Although it is possible to get close to saturating this inequality, one cannot violate it. Now, the measurement $\{A_i \otimes \hat{n}\}_i$, is only one choice of many joint measurements on system and meter (i.e. POVMs defined on the product Hilbert space). Therefore we have the following relation between the corrected Fisher information of this paper and the Quantum Fisher information $Q := \max_{\text{POVMs}} FI$ in the joint system-meter state after interaction

$$|\langle\phi|\psi\rangle|^2 FI \leq Q \left[U(|\psi\rangle\langle\psi| \otimes \sum_{\alpha} p_{\alpha} |\alpha\rangle\langle\alpha|) U^{\dagger} \right].$$

Convexity is a fundamental property of Q and also of FI . It *immediately* implies that the classical noise represented by p_{α} cannot increase the information. It is always better to choose *one* element out of the ensemble (whichever has the highest information). So in answer to question 3, it is clear that by using classical noise one is choosing to operate far below optimal precision limits – this seems a high price to pay just to get some different scaling behaviour. Next, use the convexity Q to reach

$$Q \left[U(|\psi\rangle\langle\psi| \otimes \sum_{\alpha} p_{\alpha} |\alpha\rangle\langle\alpha|) U^{\dagger} \right] \leq \sum_{\alpha} p_{\alpha} Q \left[U(|\psi\rangle\langle\psi| \otimes |\alpha\rangle\langle\alpha|) U^{\dagger} \right]$$

Now each term in the sum is proportional to the Quantum Fisher information of a pure state with no quantum correlations between photons. As the authors state, these will be $\propto N$ (corresponding to a precision, which is $\sqrt{1/Q}$, scaling $\propto 1/\sqrt{N}$). So, let each one be $A_{\alpha} N$ for some scalar proportionality constants A_{α} . Now we have

$$|\langle\phi|\psi\rangle|^2 FI \leq \sum_{\alpha} p_{\alpha} A_{\alpha} N \propto N.$$

Therefore the authors' corrected Fisher information is upper bounded by a quantity proportional to N , which is following the Standard Quantum Limit (SQL). Therefore if

$|\langle\phi|\psi\rangle|^2 FI \propto N^2$, this can only hold for a limited range of N – otherwise it would contradict the bound above. In other words it can only be a temporary – and not scalable – Heisenberg scaling, which rather undermines the whole point of having a Heisenberg scaling in the first place.

The HL scaling here is nothing more than a curiosity. Similar arguments to the above would show that by setting $|\psi\rangle$ such that $C|\psi\rangle = c_*$ (the largest eigenvalue of C), one would obtain more Fisher information (again scaling with the SQL) using a measurement of the phase of the optical beam than would be achieved with the technique in this paper. I encourage the authors to perform this calculation themselves. This does not diminish the fact that the use of imaginary weak values has enabled a photon counting measurement to be used instead of a phase sensitive measurement. This is clearly advantageous (not least because it suppresses the unwanted influence of the self-phase modulation); in my opinion it is much more important than the temporary Heisenberg scaling. Concentrating on the latter is rather missing the point of metrology. The authors should rewrite their paper with the former feature as the focus.

1.2. Presentation. It should be made clear that equation (1) is only valid for pure states. This is crucially important, and it is arguably misleading in the present manuscript since ultimately the authors will move away from pure states. Linked to this is the statement at the bottom of page 2 that ‘the area of the disk is fixed’. Again this is only true for pure states. Let me explain why this could lead to confusion. In the end, precision is controlled by the variance of a given quadrature in phase space. For pure states that are Fourier-transform limited, coherently increasing the variance in one direction necessarily involves decreasing it in a conjugate direction. This is why the Fisher information is proportional to a variance (which would otherwise be counterintuitive) for pure states. As soon as you move from pure states to mixed states, that relationship is broken. Ultimately, the authors use an unusual technique where again the precision is proportional to a variance – but this quite distinct from the well known theory behind equation (1). So repeatedly revisiting this idea is not helpful for the reader of this work. Further, equation (9) of the supplementary material – which seems to have a typo, too many \sim s – is also only true for a unitary family of pure states. I refer the authors to equations (28) and (31) of Paris, ‘Quantum Estimation for Quantum Technology’ *Int. J. Quant. Inf.* 7 125-137 (2009). This reference shows that there are extra terms in the mixed state case.

2. MINOR CONCERNS

2.1. Presentation. It is not actually clear what Figure 1 is supposed to illustrate. In panels a,b,d, what do the blue and red patches denote? What meaning should I attach to their area, or the overlap? In panel d can I visualise the shift in the radial direction? Usually, the precision should be better if the overlap between initial and final states is smaller; but it is not clear at all from this Figure if that is being claimed or shown. I think improvements are in order to assist the uninitiated reader. Why do the authors refer to a minimum width / maximum length for a squeezed state?

We are not told what p_α looks like, despite this being a very important component of the protocol. From the Figure 1, it looks like it has support on six values of α . But is it uniformly distributed? Please clarify.

The authors use the tilde symbol \sim a lot without defining it. Does it mean proportional to, or approximately equal to, or something else? If the former, may I suggest the LaTeX command `\propto`.

2.2. References. References [31,33] are cited in a very generic manner: but they are clearly very relevant to this work, explicitly mentioning Heisenberg scaling with weak values. Surely these deserve more of a discussion?

The derivations (including those in the supplement) are lacking in mathematical detail. How do I get to equation (10) of the supplementary material? Why not follow NJP 12 023036 (2010) or even Physics Reports 520 43-133 (2012) ?

Also, other papers measuring a cross phase modulation with weak value approaches would seem relevant: Nature Physics 11 905-909 (2015), and Nature Physics DOI: 10.1038/NPHYS4040 (advance online publication).

I am also surprised that the authors do not cite similar work from their own experimental group PRL 111 03360 (also using weak measurements with an imaginary weak value with an incoherent probe).

δP not defined, should be $P \rightarrow P + \delta P$ right above equation (2).

3. RECOMMENDATION

This paper is not publishable in its current form, since it has the completely unacceptable use of ‘scalable’ in the title. It also puts the wrong emphasis on Heisenberg scaling. Although this it *technically* achieved in this paper, it is at the expense of optimal precision which is after all the main point of metrology. I recommend the authors prepare a major revision, concentrating on the advantage gained by being able to measure photon number instead of phase using their technique. They must be clear in the next version that the precision of their method is always dominated by a phase sensitive technique that scales with the SQL. In Figure 4b, this would be another trace with a shallower gradient that is always below the red line.

Reviewer #3 (Remarks to the Author):

The manuscript "Scalable Heisenberg-limited metrology using mixed states" by Cheng et al. proposes a protocol for reaching the Heisenberg scaling in quantum metrology. The key result relies in a combination of a weak measurement with strong post-selection and the use of mixed states to increase the fluctuations of the system probe.

I find this manuscript interesting, and in my opinion the result -if correct- deserves publication. However, I have found this manuscript quite technical and difficult to follow even for an audience relatively close to the quantum metrology community.

First, there are several imprecisions that make confusing the reading. Namely

- Eq.(1) is only correct if H depends linearly on g , whereas no linear dependence was introduced before. Moreover, the limit in terms of the variance is not tight for mixed states. In this case, the variance must be replaced by the quantum Fisher information. In addition, the authors do not mention that ΔH must be computed in the quantum initial state.

- After Eq. (1), the authors write "By preparing an initial state with a large ΔH , one can improve the precision". This is confusing, and I understand that if we want to improve the precision, we need states with large ΔH . That is correct. However, the latter property by itself does not guarantee any success, as the completely mixed state maximizes ΔH but it is useless.

- After Eq. (1) the authors also comment that "The interaction can involve a large number, N , of subsystems. In this case $H = \sum_{i=1}^N H_i$ ". This is very confusing because if the Hamiltonian includes interactions, such a decomposition is not possible. Actually, I do not see this is the analyzed case in subsequent sections.

- The explanation of Fig 1 is also confusing. Graphically the figure is ok, but the authors does not comment why such scaling laws ($1/\sqrt{N}$, $1/N$) are obtained. Their short explanation induces to believe they are a rather trivial consequence of the figures, and I do not see that.

- Concerning the new reported results, a more detailed derivation of Eqs. (4) and (5) would be appreciated.

Secondly, the experimental analysis seems consistent, but authors need to make the approximation $N g \ll 1$, which implies that the Heisenberg limit cannot be obtained in the form of $1/N_{\text{total}}$, being $N_{\text{total}} = N \nu$. This is the true limit imposed by quantum mechanics as demonstrated recently for instance in “V. Giovannetti et al. ‘Quantum Measurement Bounds beyond the Uncertainty Relations’, Phys. Rev. Lett. 108, 260405 (2012)” and “M. J. W. Hall and H. M. Wiseman ‘Heisenberg-style bounds for arbitrary estimates of shift parameters including prior information’, New J. Phys. 14, 033040 (2012)”. There is no practical limit in terms of Eq. (1) as ΔH can be arbitrarily high keeping finite the mean number of photons (see the references above).

Anyhow, the fact that the authors can realize precise measures reaching the Heisenberg scaling (instead of limit) by exploiting mixedness is interesting, and after a major revision the paper can be suitable for Nature Communications.

Incidentally, I think the first reference to suggest that mixed states can outperform coherent states in the case of nonlinear optical couplings is “A. Rivas and A. Luis, ‘Precision Quantum Metrology and Nonclassicality in Linear and Nonlinear Detection Schemes’ Phys. Rev. Lett. 105, 010403 (2010)”.

Reviewer #4 (Remarks to the Author):

Comments for Author

The author presented and experimentally implemented a measurement scheme that utilizes the weak value. The major claim is that their scheme achieves the Heisenberg limit with only mixed initial states.

I cannot recommend this manuscript to publish in Nature Communications, because I think the scheme presented by the authors does not achieve the Heisenberg-limited metrology. The reasons are given as follows:

1. In the Abstract, the authors mentioned the attempts of using NOON states and squeezed states to achieve the Heisenberg limit, and then said they presented and implemented a scheme enables the Heisenberg limit to be reached. However, the "Heisenberg limit" used in the manuscript is not as the same as that in the works of using NOON states and squeezed states.

The authors considered a very restrictive model where the parameter to be estimated is equivalent to the strength of the coupling between a system and a probe, a post-selection is performed on the system, and the same coupling operator is measured on the probe. Furthermore, the coupling is assumed to be weak such that the measurement on the probe after the post-selection gives the imaginary part of the weak value of the coupling operator of the system. Based on this very restrictive model, the author derives a scaling behavior $1/N$ of the estimation error regarding some characteristic quantity N and named this scaling as the Heisenberg limit. However, this is different with the Heisenberg limit in the common sense, where the sensing operation is linear and independent and all kinds of measurements are considered. This can be convinced by looking at the most of the references of the manuscript. Although the scheme given in the manuscript is interesting, claiming that it achieves the Heisenberg limit, I think, is confusing and inappropriate.

2. In this work, only the post-selected events are taken into account. This is another reason that makes the claim of achieving Heisenberg limit questionable. The application of post-selection and weak value for metrology has its own significance, however, when claiming a work based on post-selection achieving the Heisenberg-limit, I believe more rigor and systematic analysis on the resource counting is necessary to make the claim convincing to the quantum metrology community.

I also have the following technical points.

3. To derive the lower bound Eq. (1), the parameter g should be moved out of H . That is, Eq. (1) is valid for a unitary sensing $U = \exp(-igH)$ but not $U = \exp(-iH(g))$.

4. Below Eq. (2), the authors said, "calculating the signal to noise ratio, ..., yield a precision $\Delta g \sim \Delta P^{-1}$ ". I cannot say how this precision can be connected to the signal to noise ratio. Actually, as the author presented in the Supplementary Material, the precision, or estimation error Δg is defined through error propagation but not the signal to noise ratio.

5. It seems that the author used the variance of the Hamiltonian as the quantum Fisher information up to a constant multiplicative factor, see Eq. (9) of the Supplementary Material. However, this

relation is only valid for pure states. Since the authors consider mixed probe states or classical fluctuations, I cannot see how the quantum Fisher information can be obtained from the variance of the Hamiltonian. The author should clarify the meaning of Eq. (9) and its relation to quantum Fisher information.

Article reference: NCOMMS-17-08474

Title: Scalable Heisenberg-limited metrology using mixed states

Authors: Geng Chen, Nati Aharon, Yong-Nan Sun, Zi-Huai Zhang, Wen-Hao Zhang, De-Yong He, Jian-Shun Tang, Xiao-Ye Xu, Yaron Kedem, Chuan-Feng Li, and Guang-Can Guo.

Reply to referee 1

Comment 1.1: *“I have read thoroughly the manuscript entitled “Scalable Heisenberg-limited metrology using mixed states”, by Prof Li and co-workers. The paper describes an experiment in quantum metrology using post-selection, where the cross-Kerr nonlinearity between an intense probe field and a single photon is measured. The authors measure the shift in the mean photon number of the probe state, conditioned on a post-selection of the single-photon state. The authors demonstrate that the precision of the scheme scales as $1/N$, where N is the average photon number of the probe field. This is known as Heisenberg-like scaling. This result is rather surprising, since the experiment is performed using a probe that is a mixture of coherent states. A coherent state by itself would give a precision that scales as $1/\sqrt{N}$, known as the standard quantum limit (SQL). For this reason I believe that most physicists in the field would naturally assume that a mixture of coherent states would also follow the SQL. Thus, I found the overall result to be quite interesting. The result seems to contradict the following intuition: The Fisher information is convex, so the Fisher information available using the ensemble of coherent states is less than or equal to the weighted sum of the Fisher information of each coherent state. Since each coherent state obeys the SQL, how is the Heisenberg scaling reached? Moreover, the authors argue that the classical Fisher information of the scheme is close to the quantum Fisher information by less than an order of magnitude. The theory provided by the authors appears to be correct, so what is wrong with the above reasoning?”*

Response: We are glad that the referee found our work interesting. Reading the reports of the referees, we have realized that there is some confusion regarding the achieved precision and its scaling. We would like to stress that the precision is always bounded by the quantum Fisher information (QFI) of the *joint* system-probe state. Let’s consider first a pure state of the probe (always assuming a pure state of the system), the QFI of the joint system-probe state after the interaction is $\propto N^2$, rather than $\propto N$, and this is the crucial observation here. However, if the initial state of the probe is a coherent state, then the obtained precision by measuring only the probe is limited by the SQL, as mentioned by the referee. For mixed probe states, the QFI is bounded by the weighted sum of the QFI of the pure joint states, which is also $\propto N^2$. Interestingly, our scheme results in a (classical) FI that is $\propto N^2$ and which is obtained by measuring the probe. In the usual method (measurement of a coherent state probe with no post-selection) the precision is SQL limited, and because mixed states are utilized, our scheme is Heisenberg scaling, which makes it counter-intuitive. However, it does follow the limit set by the QFI.

Changes made: We have clarified this issue both in the main text and in the supplementary. In the main text, on page 2 below the Fig.1, we have added a new paragraph on the QFI and the obtained FI in our scheme. In the supplementary we have revised the sections on the quantum and classical FI (sections I and II respectively), where a detailed derivation is provided. In addition, the summary has also been revised along these lines.

Comment 1.2: *“I admit that it took some time for me to believe that this result is indeed possible. I would say that the single photon plays an extremely important role here. For example, purity (coherence) of the system photon is required for the scheme to work. It is also necessary that the parameter to be estimated (g) appear in an interaction Hamiltonian between the system and probe, which allows one to take advantage of the coherence of the system. In this regard, I believe that the authors do not emphasize the role of the system photon sufficiently. For example, the title and abstract only mention the mixed probe states. The abstract should mention that the “post-selection” is indeed “post-selection of an additional pure system”, or something along these lines. Surely adding technical noise to a standard experiment would be detrimental. However, with this interaction Hamiltonian and an additional pure system photon, the classical noise becomes an asset. What additional resource is provided by the system photon?”*

Response: The purity of the system is indeed crucial for our scheme to work properly. A QFI of $\propto N^2$ is possible only due to a coherent interaction between the system and the probe. The added technical noise, together with the specific post-selection (an imaginary weak value) allow to obtain this scaling by a measurement of the probe. We agree with the referee that this hasn’t been stressed enough in the manuscript and we have revised it accordingly.

Changes made: We have revised the abstract significantly, where the role of the pure quantum system is stated explicitly. We have also revised the summary where now it begins with “In summary, in this work a new scheme for the metrological task of estimating a weak coupling strength between a pure quantum system and a probe was presented. We theoretically and experimentally demonstrated that mixed probe state, combined with a post-selection of the pure quantum system, can be utilized to improve the precision. Specifically, the extracted FI is close to and scales as the QFI.”

Comment 1.3: *“The experimental is very nice, as are the experimental results. However, the authors should provide a little more discussion in the text concerning their main results in Fig. 4. Do the linear fits in (a) have slope proportional to N ? What is the $1/N$ curve fit in Fig. (b)? How does this compare to the precision limit given by the quantum Fisher information?”*

Response: The linear fitting slope for previous Fig. 4(a) is nearly proportional to N and the exact values are cited in Fig. S2 in the supplementary. The $1/N$ curve fit describes a Heisenberg scaling behaviour. In the revised manuscript, we have changed the linear coordinates to logarithmic coordinates. As a result, the Heisenberg scaling behaves as a linear line as shown in the new Fig. 4, and the coefficient is given in the caption to be $6.3 * 10^{-4}$. We also show the precision limit calculated from the bound of QFI for mixed state (the red line in new Fig. 4). It can be seen that the precision achieved in our experiment is in the same order of magnitude with the precision limit.

Changes made: We have redrawn Fig. 4 with new experimental data for larger values of N and we have changed the axis to logarithmic coordinates. A theoretical precision limit is plotted as the red line in the new Fig. 4 and we have modified the caption accordingly. We have moved previous Fig. 4(a) (together with new plots) to the supplementary, where we elaborate on the quantification of fitting slopes to calculate the precision.

Comment 1.4: *“There are several conditions for the scheme to work: $\epsilon \ll 1$; $Ng \ll 1$; $ng/\epsilon \ll 1$. It would be interesting if the authors would briefly discuss the theoretical*

limit to the average photon number that could be used, in the interest of optimizing precision. Do the authors perform the experiment near this value? ”

Response: In order to answer this question and show how long can the Heisenberg scaling survive, we have performed new experiments with larger values of N up to 5×10^6 . As shown in the new Fig. 4, the precision deviates from the green line (Heisenberg scaling) when N is larger than 10^5 , i.e., when $Ng/\epsilon \simeq 0.1$. We have further made numerical simulations of the precision for varying values of Ng/ϵ as shown in the supplementary (section III. A). In agreement with the experimental results, our theoretical analysis indicates a deviation from the HS when $Ng/\epsilon \simeq 0.1$, and an ultimate precision appears when $Ng/\epsilon \simeq 10$. However, due to the leakage from the probe pulses in the case of large N , this optimal precision has not been achieved experimentally.

Changes made: We have modified Fig. 4, where the new experimental data is now shown. In the supplementary, we have added a section on the small parameter regime (section III. A). In addition, we have added a section on the quantification of the slopes (section V), where the original and new plots are shown (Fig. S2).

Comment 1.5: *“In summary, I find the result to be very interesting, but I believe that many readers will struggle to understand why this result is possible. The theory provided in the manuscript is more or less straightforward, but the argument based on simply increasing the variance using classical noise seems to describe something that is too good to be true. Could the authors give a more intuitive argument as to why their scheme achieves Heisenberg scaling? There must be some quantum resource (entanglement? discord? coherence? squeezing?) responsible for surpassing the SQL. What is it? ”*

Response: We hope that our reply to the above comments of the referee makes this point clear. Again, due to the coherent interaction between the system and the probe, the QFI of the joint system-probe state is $\propto N^2$. Our scheme enables to extract an FI which also scales as $\propto N^2$ by a measurement of a probe with a large classical variance, conditioned on a specific post-selection of the system. The combination of the large classical variance and the post-selection make the FI be extractable by a measurement of the probe.

Changes made: See comments 1.1 to 1.4.

Comment 1.6: *“I believe that the manuscript describes an important and counter-intuitive result that could appear in Nat. Comm, provided the authors can address my comments above. The paper is well-written, as is the supplementary material. Below I mention some additional minor corrections that should be made. ”*

Response: We believe that we have answered all comments raised by the referee in a satisfactory manner. We have made a significant revision of the manuscript and the supplementary accordingly. Thanks to the comments of the referee our manuscript is now greatly improved and we believe that it is suitable for publication in Nat. Comm.

Comment 1.7: *“Minor corrections: Page 2: Why is the probe operator \hat{n} written in operator notation, and the operator C is not? Page 2, second column: I suggest changing “...we measure it’s photon number” to “... we measure it’s average photon number”. Methods section: There are a few typos: “secondary harmonic generation” \rightarrow “second harmonic generation” ; “cutting angle” \rightarrow “cut angle”, “duplet lens” \rightarrow “doublet lens”. I also*

believe that "the 785 photon is reflection ... as a pump" pulse \rightarrow "the 785 photon is reflection ... as a system pulse." Detection apparatus section: The post-selection state should be φ and not ψ "

Response: We thank the referee for bringing these corrections to our attention. All minor corrections have been made.

Reply to referee 2

We are truly grateful for the comprehensive and detailed report of referee 2. We believe that the report has helped us to improve the manuscript considerably. Before we go over the report point by point, we would like to clarify one crucial issue.

The referee derives a bound for the precision using the Quantum Fisher Information (QFI) of the joint system-probe state after the interaction but before the post-selection, or any other final measurement. At the end of page 2 in the report, just above the last equation (equation. (4) on page 8 below), the referee writes that “As the authors state, these will be $\propto N$ (corresponding to a precision, which is $\sqrt{1/Q}$, scaling $1/\sqrt{N}$.” However, the QFI of these (joint) states is actually $\propto N^2$, as can be seen clearly in the new version of the manuscript and the supplementary material. Hence, in the last equation on page 2 of the report the bound is proportional to N^2 and does not exclude Heisenberg scaling. The point we were trying to make was that using the standard method (measurement of a coherent state probe with no post-selection), would yield a precision $\propto 1/\sqrt{N}$. Interestingly, in our scheme a measurement of the probe results in a precision $\propto 1/N$. The achieved precision in this work is close to the optimal bound set by the bound of QFI, which is approximately $1.2N^2$ in our case.

Naturally, this confusion has led us to revise the manuscript considerably. We suspect that the caption of Fig. 1 has contributed much to this mix-up and thus it has been completely rewritten. A few paragraphs in the main text have been added and some paragraphs have been rewritten. A special section in the supplementary material is now dedicated to this point. A full description of all the changes is given below, as well as in the highlighted version of the manuscript, where all changes made are highlighted.

Comment 2.1: *“This paper deals with statistical estimation of g – the strength of a cross-phase modulation in an optical experiment – claiming scalable Heisenberg-limited measurement precision using only mixed states. If the theory and experiment for this paper stand up to scrutiny, it would seem to be a very significant result, since usually it is believed that a resource such as entanglement is necessary for such a quantum advantage (where the precision scales inverse-linearly rather than inverse-root with the number of photons N). Furthermore, mixed states are generated with classical randomness which is widely believed to reduce quantumness and its associated advantages (whereas here it appears to improve things). Also, the technique presented here is based in the formalism of weak measurement. This suffers from lots of controversy, mainly because of the necessary use of postselection (discarding of data or experimental runs). Postselection is known to be a process that can obscure the true physics and easily lead to nonsensical or misjudged reasoning. It seems then that there is much to be sceptical of in this paper; but seeing as there is also much at stake, let us take a systematic and fair approach to assessing it.”*

Response: We agree with the referee regarding the counter-intuitive nature of our work. We hope that in light of the clarifications and major modifications that we have presented in the revised manuscript, she/he will agree that it also stands up to scrutiny. As we note above, the QFI of the joint system-probe state after the interaction is $\propto N^2$. However, while in the usual method (a phase measurement of a coherent state probe with no post-selection) the precision is SQL limited, our scheme enables to extract Fisher information (FI) that is also $\propto N^2$ by a measurement of the probe. Hence, a precision with Heisenberg scaling is achieved. This is one aspect of the counter-intuitive nature of our work. The second counter-intuitive aspect is the benefit of mixed states. As noted by the referee, mixed states correspond to classical randomness that usually reduce quantumness and its

associated advantages. In our scheme, however, the precision is improved by increasing Δn also classically, and hence, in such a set-up, the classical randomness of mixed states does improve the precision.

Indeed, weak measurements are a controversial topic, usually with respect to the so called "weak value amplification" and its possible advantages in precision measurements. We are happy that the referee, while still acknowledging the controversy, takes a rather unbiased approach. We would like to note that our main result does not depend on such an "amplification" (a large weak value). With this respect, the key element in our scheme is an imaginary weak value. We have aimed to present our results in a plain and clear way, without the hype that is sometimes attributed to weak measurements. In addition, please note that the precision is independent of the probability of post-selection. In the expression of classical FI (Eq. S13), the post-selection probability is compensated by the square of weak value. This is discussed in the main text (second paragraph after Eq. (5)) and in the revised supplementary (section III. B).

It should be emphasized that in addition to the theoretical derivations, our work includes an experimental demonstration reaching an unprecedented precision in a measurement of the Kerr phase of a single photon, which is also an additional support to the validity and feasibility of the theory.

Changes made: Please see all comments below.

Comment 2.2: *"1.1. The main argument. This work is based on theoretical results published in [23], and can be thought of as a protocol which makes two choices: i) to use pre and postselection to arrange a shift in n (by a quantity proportional to $g\Delta n$, where Δn is the uncertainty in photon number) rather than a rotation of the phase (by a quantity proportional to g) and ii) use of classical noise to increase Δn . "*

Response: In general, this is a faithful description of the technique. The shift in our case is proportional to $g\Delta n^2$, but this is probably just a typo.

Comment 2.3: *"With all this in mind, in assessing this paper I believe there are at least the following 4 key questions: 1) Does the proposed technique really offer Heisenberg scaling when all resources are taken into account? 2) Does it continue to do so when N is taken arbitrarily high? In other words is it really a scalable technique? 3) Is the classical noise playing a crucial role? 4) Are there theoretically superior techniques that could have been just as easily used? "*

Response: Indeed, these questions are central in assessing our work. We give our response below following each answer of the referee.

Comment 2.4: *"In response to question 1: I have checked the theoretical proofs from Ref [23] and I find that they are correct in a certain parameter regime. Therefore we can say that there is indeed a Heisenberg scaling with this technique. "*

Response: Heisenberg scaling (question 1) — We appreciate the time invested and the efforts made by the referee in order to evaluate our work and confirming that a Heisenberg scaling is indeed achieved.

Comment 2.5: *"However, the scaling will not continue forever and therefore we must crucially conclude question 2 in the negative. This is because, as the authors state, their method requires $Ng \ll 1$. for any fixed value of the unknown parameter g , there is a maximum*

N such that this condition is satisfied. See Figure 1 of Ref [23], where the signal-to-noise ratio (roughly equivalent to Fisher information) actually starts to decrease when the technical noise (equivalent here to Δn) is increased beyond a certain value. The authors will argue (correctly) that the temporary scaling can survive for longer and longer (for higher and higher values of Δn) as long as g gets smaller and smaller. But since in any real experiment g is finite it is clear that the scaling will always be temporary.”

Response: Scalability (question 2) — Indeed, the Heisenberg scaling is achieved in the regime where $Ng/\epsilon \ll 1$. When we termed our scheme “scalable” we had a practical notion of scalability in mind. Since for all known experimental results with techniques such as NOON states, the photon number is on the order of 10 (or less), we thought that demonstrating our method using 10^4 to 10^5 photons would justify the term “scalable” from a practical point of view. The referee is of course correct that since $Ng/\epsilon \ll 1$ is required in our scheme for achieving the Heisenberg scaling, our scheme is not theoretically scalable in the sense that for a given value of g there will always be a large enough value of N for which $Ng/\epsilon \ll 1$ does not hold. In order to avoid confusion or misleading, we have decided not to use “scalability” with respect to our scheme. However, the practical notion of scalability is of great importance because, practically, no technique is infinitely scalable. Technical issues would eventually limit the amount of resources, e.g., the number of photons, and in some cases to even lower values than the limit posed by our approximation. As the referee points out, we think that a most interesting scenario is of a vanishing g and in the revised version of the supplementary we analyze the scaling behaviour of the scheme based on taking smaller and smaller values of g .

Changes made: The word “scalable” has been removed from the title. We have carried out further experiments and obtained some new data. In the revised version of the manuscript experimental results up to $N = 5 * 10^6$ are presented and Fig. 4 has been extended up to $N = 5 * 10^6$. The regime where $gN/\epsilon \simeq 1$ is investigated also numerically and the deviation from the Heisenberg scaling is shown both theoretically and experimentally. Regarding the small parameter regime (vanishing g), we have defined a scaling behaviour in the last paragraph before the summary (page 4, below Fig. 4), which is explained in detail in the supplementary section III. A.

Comment 2.6: “To answer questions 3 and 4, we will proceed by considering an alternative measurement strategy. Consider that the postselected state is but one outcome of a set of operators A_i resolving the identity $\sum_i A_i = \mathbb{I}$. For example, $A_1 = |\phi\rangle\langle\phi|$. The Fisher information associated with the entire (unconditioned) resultant probability distribution will always be greater than or equal to the Fisher information conditioned of one particular outcome, when it has been corrected for the probability of obtaining that outcome:

$$|\langle\phi|\psi\rangle|^2 \nu FI \leq \nu FI \left[\text{Tr} \left((A_i \otimes \hat{n}) U \left(|\psi\rangle\langle\psi| \otimes \sum_{\alpha} p_{\alpha} |\alpha\rangle\langle\alpha| \right) U^{\dagger} \right) \right]. \quad (1)$$

Although it is possible to get close to saturating this inequality, one cannot violate it. Now, the measurement $\{A_i \otimes \hat{n}\}_i$, is only one choice of many joint measurements on system and meter (i.e. POVMs defined on the product Hilbert space). Therefore we have the following relation between the corrected Fisher information of this paper and the Quantum Fisher information $Q := \max_{\text{POVMs}} FI$ in the joint system-meter state after interaction

$$|\langle\phi|\psi\rangle|^2 FI \leq Q \left[U \left(|\psi\rangle\langle\psi| \otimes \sum_{\alpha} p_{\alpha} |\alpha\rangle\langle\alpha| \right) U^{\dagger} \right]. \quad (2)$$

Convexity is a fundamental property of Q and also of FI. It immediately implies that the classical noise represented by p_α cannot increase the information. It is always better to choose one element out of the ensemble (whichever has the highest information). So in answer to question 3, it is clear that by using classical noise one is choosing to operate far below optimal precision limits – this seems a high price to pay just to get some different scaling behaviour. Next, use the convexity Q to reach

$$Q \left[U \left(|\psi\rangle\langle\psi| \otimes \sum_{\alpha} p_{\alpha} |\alpha\rangle\langle\alpha| \right) U^{\dagger} \right] \leq \sum_{\alpha} p_{\alpha} Q \left[U \left(|\psi\rangle\langle\psi| \otimes |\alpha\rangle\langle\alpha| \right) U^{\dagger} \right]. \quad (3)$$

Now each term in the sum is proportional to the Quantum Fisher information of a pure state with no quantum correlations between photons. As the authors state, these will be $\propto N$ (corresponding to a precision, which is $\sqrt{1/Q}$, scaling $\propto 1/\sqrt{N}$). So, let each one be $A_{\alpha}N$ for some scalar proportionality constants A_{α} . Now we have

$$|\langle\phi|\psi\rangle|^2 FI \leq \sum_{\alpha} p_{\alpha} A_{\alpha} N \propto N. \quad (4)$$

Therefore the authors' corrected Fisher information is upper bounded by a quantity proportional to N , which is following the Standard Quantum Limit (SQL). Therefore if $|\langle\phi|\psi\rangle|^2 \propto N^2$, this can only hold for a limited range of N – otherwise it would contradict the bound above. In other words it can only be a temporary – and not scalable – Heisenberg scaling, which rather undermines the whole point of having a Heisenberg scaling in the first place. The HL scaling here is nothing more than a curiosity. Similar arguments to the above would show that by setting $|\psi\rangle$ such that $C|\psi\rangle = c_*$ (the largest eigenvalue of C), one would obtain more Fisher information (again scaling with the SQL) using a measurement of the phase of the optical beam than would be achieved with the technique in this paper. I encourage the authors to perform this calculation themselves. ”

Response: Crucial role of classical noise (question 3) — The derivation of the referee up to (and including) equation (3) is indeed correct. However, as we show in the revised manuscript (and supplementary) in the case of an initial pure probe state, the QFI of the joint system-probe state after the interaction is $\propto N^2$, and in this case equation (4) actually reads $|\langle\phi|\psi\rangle|^2 FI \leq \sum_{\alpha} p_{\alpha} A_{\alpha} N^2 \propto N^2$. Hence, there is no contradiction with the Heisenberg scaling. However, in the regime $Ng/\epsilon \ll 1$ the usual phase measurement of the probe can extract an FI of only $\propto N$, and the precision is therefore limited by the SQL. In our scheme the combination of an imaginary weak value *and* classical noise, which increases Δn , enables us to extract an FI of $\propto N^2$ by a measurement of the probe. Using only an imaginary weak value, without the classical noise would yield an FI of $\propto N$ and an SQL limited precision. The convexity of the FI seems to imply that a classical noise cannot improve the precision (which is of course correct in the usual sense mentioned by the referee). The (corrected) equation (4), $|\langle\phi|\psi\rangle|^2 FI \leq \sum_{\alpha} p_{\alpha} A_{\alpha} N^2 \propto N^2$, indeed suggests that the QFI in the case of a mixed probe state cannot be greater than the QFI due to one optimal pure probe state. However, it would be impossible to extract an FI $\propto N^2$ by a usual measurement of the probe. Our scheme does enable to do just that! In such a set-up, the classical noise has a crucial role.

Theoretically it is always better to choose the best member of the ensemble, which results in the largest QFI. However, there are a few issues at hand that we would like to point out. First, one has to consider the efficiency of the final measurement; having a final state with an optimal QFI does not guarantee a simple method to extract the information. In the scenario considered in this work, in the standard method of a phase measurement

of the probe the extracted FI is $\propto N$ and the precision follows the SQL. In our scheme the extracted FI is $\propto N^2$ and the precision follows the HS. Second, one should note that precision measurements are, in the end, a practical task and thus, practical aspects should be considered. In our scheme the detrimental effect of the self-phase modulation is eliminated while the HS is achieved by a measurement of the probe. We have achieved an unprecedented precision in a measurement of the Kerr phase of a single photon. To the best of our knowledge, the achieved precision for this task in previous experimental works, where pure coherent states have been utilized, was SQL limited. Third, since the average photon number per pulse in our method is N , it should be compared to a pure coherent state with the same number of photons, N , and not to a pure state with the maximal photon number from the distribution. This makes more sense in terms of resource counting, and in this case, the achieved precision in our scheme is lower only by a constant factor ($\simeq 5$) compared to the optimal bound (due to a pure state) set by the QFI. Moreover, in principle, using a probe state with a maximum variance ($\sim N^2$) we can attain this optimal bound.

We would like to stress that the precision is independent of the post-selection probability, which is given by the factor $|\langle\phi|\psi\rangle|^2$, and does not result due to any miscounting of the post-selected resources. We emphasize that this is not the case. As explained now in the supplementary material section III. B, when counting the total information carefully this factor cancels out with the pre-factor due to the weak value.

All in all, we believe the answer to question 3 “Is the classical noise playing a crucial role?”, should be: It is not crucial from a theoretical point of view but it is vital for the practical method we used.

Possibility of theoretically superior techniques (question 4) — Equation (S1) in the revised version of the supplementary, shows that for large N , setting the system to an eigenstate of \hat{C} would diminish the amount of information considerably; the QFI would be $\propto N$, rather than $\propto N^2$. Moreover, the technique suggested by the referee (phase measurement of the probe beam) is essentially the one used in Ref. [25] (revised manuscript), which our experiment outperforms by two orders of magnitudes. Again, while in Ref. [25] the precision follows the SQL, in our scheme it follows the Heisenberg scaling. There may be some other techniques that, theoretically, can achieve a nearly same precision, however, to the best of our knowledge our method is the first one to achieve the Heisenberg scaling experimentally by utilizing a mixed probe state, which is commonly regarded to be useless for improving precision. We hope that our scheme will prove to be useful in other scenarios as well.

Changes made: Regarding the obtained precision, we have clarified this issue both in the main text and in the supplementary. In the main text, at the end of page 1 we have added a new paragraph on the QFI and the obtained FI in our scheme. In the supplementary we have revised the sections on the quantum and classical FI (sections I and II respectively), where a detailed derivation is provided. In particular, the QFI for a pure state is shown explicitly in section I of the supplementary, equation (S1). The caption of Fig. 1 was rewritten in order to explain more clearly how is the Heisenberg scaling achieved. In addition, the summary has also been revised along these lines.

Comment 2.7: *“This does not diminish the fact that the use of imaginary weak values has enabled a photon counting measurement to be used instead of a phase sensitive measurement. This is clearly advantageous (not least because it suppresses the unwanted influence of the self-phase modulation); in my opinion it is much more important than the temporary Heisenberg scaling. Concentrating on the latter is rather missing the point of*

metrology. The authors should rewrite their paper with the former feature as the focus.”

Response: The elimination of the self-phase modulation was initially a major motivation for this work. Indeed, we agree that it is an important effect on its own. However, the fact the overall precision (truly) scales as $1/N$ seems to be even more significant and interesting to a general audience.

Changes made: We have emphasized the role of the imaginary weak value along the manuscript. In the abstract a sentence highlighting this effect has been added. In the main text a new paragraph, just below Eq. (5) has been added, which is dedicated to this point. Another sentence has been added to the discussion, further emphasizing this issue.

Comment 2.8: *“1.2. Presentation. It should be made clear that equation (1) is only valid for pure states. This is crucially important, and it is arguably misleading in the present manuscript since ultimately the authors will move away from pure states. Linked to this is the statement at the bottom of page 2 that ‘the area of the disk is fixed’. Again this is only true for pure states. Let me explain why this could lead to confusion. In the end, precision is controlled by the variance of a given quadrature in phase space. For pure states that are Fourier- transform limited, coherently increasing the variance in one direction necessarily involves decreasing it in a conjugate direction. This is why the Fisher information is proportional to a variance (which would otherwise be counterintuitive) for pure states. As soon as you move from pure states to mixed states, that relationship is broken. Ultimately, the authors use an unusual technique where again the precision is proportional to a variance – but this quite distinct from the well known theory behind equation (1). So repeatedly revisiting this idea is not helpful for the reader of this work. Further, equation (9) of the supplementary material – which seems to have a typo, too many $\sim s$ – is also only true for a unitary family of pure states. I refer the authors to equations (28) and (31) of Paris, ‘Quantum Estimation for Quantum Technology’ *Int. J. Quant. Inf.* 7 125-137 (2009). This reference shows that there are extra terms in the mixed state case.”*

Response: We agree with the referee that this hasn’t been presented clearly in the manuscript and we have revised it accordingly. Hopefully, the revised manuscript does not suffer from the warranted problems raised by the referee. We would like to emphasize that the only purpose of Eq. (1) and Fig. 1 is to help in developing some (new) intuition to the somewhat counter-intuitive and perhaps surprising result, which is derived independently. The relation between the QFI of a pure state, and the uncertainty of the measurement observable can be understood in terms of a Fourier transform to the conjugate variable. In our case, the classical Fisher information is proportional to the variance of the measured variable even when the distribution is not Fourier transform limited, i.e., when one uses a statistical mixture. Dismissing this phenomenon as just a coincidence might impede understanding the new result. We agree, however, that the above relationship between the variances of conjugated variables of pure states does not hold for mixed states. Indeed, Eq. (9) (which corresponds to Eq. S1 in the revised supplementary) is valid only for pure states. Due to the convexity of the QFI, a bound for mixed states can be derived, as is shown in the revised supplementary (section I).

Changes made: The introduction has been revised considerably, and specifically, Eq. (1) has been modified. The caption of Fig. 1 has been rewritten, as well as the discussion regarding the figure in the main text. The discussion on the QFI in the supplementary has been revised. In particular the QFI for pure states is shown and, by convexity, a bound for mixed states is derived.

Comment 2.9: “2.1. Presentation. It is not actually clear what Figure 1 is supposed to illustrate. In panels a,b,d, what do the blue and red patches denote? What meaning should I attach to their area, or the overlap? In panel d can I visualise the shift in the radial direction? Usually, the precision should be better if the overlap between initial and final states is smaller; but it is not clear at all from this Figure if that is being claimed or shown. I think improvements are in order to assist the uninitiated reader. Why do the authors refer to a minimum width / maximum length for a squeezed state?”

Response: Red and blue patches represent states before and after the interaction took place respectively. The area of the patches represents the uncertainty of the states and the overlap between two patches corresponds to their indistinguishability. In panel (d) the red and blue patches also represent states before and after the interaction takes place respectively, but before the post-selection. We indicate that the shift (for the post-selected ensemble) is in the radial direction, however, we could not think of how to visualise this better. Since we wanted to use a somewhat unified framework, we show in all three panels the states before and after that the interaction occurs. For panels (a) and (b) the overlap is proportional to the precision, since in (d) post-selection is also involved, this is not just so, and one has to consider the shifted post-selected ensemble. The general reasoning beyond Fig 1. is as follows: panel (a) depicts the general method for a phase measurement using a coherent state probe. In panel (b) the precision of the same task as in (a) is improved by utilizing a squeezed state. Panel (c) describes a specific set-up for which both (a) and (b) apply. However, for such a set-up panel (d) describes an alternative method for improving precision in which mixed probe states are utilized. Regarding squeezed states, our meaning was to say that the uncertainty in any quadrature should not be larger than its mean value. Even though this might be possible in principle, it is impractical in the context of the task at hand. In the revised manuscript the sentence on the minimum width / maximum length has been removed. We hope that all of these issues are made clear in the revised manuscript.

Changes made: Fig 1. has been modified, its caption has been completely rewritten, and the discussion regarding the figure in the main text has been revised as well.

Comment 2.10: “We are not told what p_α looks like, despite this being a very important component of the protocol. From the Figure 1, it looks like it has support on six values of α . But is it uniformly distributed? Please clarify.”

Response: The specific form of p_α does not matter at all, only the first and second moments of the distribution (mean and variance) are of importance. Fig. 1 (d) is just a simple illustration of the mixed states under consideration and should not be taken literally. In the Methods section we specify how p_α was realized in the experiment. The amplitude of the laser is modulated by an AOM according to $V = V_0(1 - D \sin(\omega t))$. In this case the average photon number, N , is determined by V_0 , and the standard deviation is decided by D . This is also stated in the main text, in the paragraph in which Fig. 3 is discussed.

Changes made: At the end of the paragraph after Eq. (3) it is explicitly stated that the particular form of the distribution does not matter. This is shown in the supplementary section II.

Comment 2.11: “The authors use the tilde symbol \sim a lot without defining it. Does it mean proportional to, or approximately equal to, or something else? If the former, may I suggest the LaTeX command `\propto`. ”

Response: We agree with the referee.

Changes made: The suggestion of the referee to use `\propto` has been implemented throughout the revised manuscript and supplementary.

Comment 2.12: *“2.2. References. References [31,33] are cited in a very generic manner: but they are clearly very relevant to this work, explicitly mentioning Heisenberg scaling with weak values. Surely these deserve more of a discussion?”*

Changes made: A citation to these references (references [27] and [28] in the revised manuscript) has been added in the first paragraph of page 2, just below Fig. 1, showing the relevance of their results to ours.

Comment 2.13: *“The derivations (including those in the supplement) are lacking in mathematical detail. How do I get to equation (10) of the supplementary material? Why not follow NJP 12 023036 (2010) or even Physics Reports 520 43-133 (2012) ?”*

Changes made: The supplementary material was rewritten in much more detail. Specifically, equation (10), which is now (S8), is now derived explicitly in section II, where also equations (4) and (5) of the main text are derived in more detail. These two papers are cited as references [19] and [20] in the revised manuscript.

Comment 2.14: *“Also, other papers measuring a cross phase modulation with weak value approaches would seem relevant: Nature Physics 11 905-909 (2015), and Nature Physics DOI: 10.1038/NPHYS4040 (advance online publication). I am also surprised that the authors do not cite similar work from their own experimental group PRL 111 03360 (also using weak measurements with an imaginary weak value with an incoherent probe).”*

Changes made: We thank the referee for bringing this to our attention. A citation to the Nature Physics papers has been added as references [32] and [33] when talking about the methods to measure single photon nonlinearity, and the PRL paper is cited as reference [21].

Comment 2.15: *“ δP not defined, should be $P \rightarrow P + \delta P$ right above equation (2).”*

Changes made: We have changed this sentence accordingly (just before equation (2)).

Comment 2.16: *“3. Recommendation This paper is not publishable in its current form, since it has the completely unacceptable use of ‘scalable’ in the title. It also puts the wrong emphasis on Heisenberg scaling. Although this is *technically* achieved in this paper, it is at the expense of optimal precision which is after all the main point of metrology. I recommend the authors prepare a major revision, concentrating on the advantage gained by being able to measure photon number instead of phase using their technique. They must be clear in the next version that the precision of their method is always dominated by a phase sensitive technique that scales with the SQL. In Figure 4b, this would be another trace with a shallower gradient that is always below the red line.”*

Response: The word “scalable” has been removed from the title (comment 2.4). Regarding the Heisenberg scaling, we hope that we have convinced the referee that an Heisenberg scaling is indeed achieved in our scheme (comments 2.1, 2.2, 2.5 and 2.6). We would like to emphasize again that the precision achieved in our experiment beats the SQL by two orders of magnitude. A plot of the SQL would be completely out of the range of Fig. 4 (in fact it will just hit the top right corner). Because the immense improvement enabled by our scheme, compared to a recent experiment (reference [25]) aimed at the same task, cannot be explained solely by avoiding the noise related to self-phase modulation, we believe that the Heisenberg scaling should still be emphasized. Nevertheless, we agree

with the referee about the importance of the imaginary weak value in our scheme and the advantages it provides. In the revised manuscript we have therefore emphasized it much more (comment 2.6). Thanks to the comments raised by the referee, we believe that the revised manuscript stands up to scrutiny and that the achievements presented in this work warrant publication in Nature Communications.

Reply to referee 3

Comment 3.1: *“The manuscript ”Scalable Heisenberg-limited metrology using mixed states” by Cheng et al. proposes a protocol for reaching the Heisenberg scaling in quantum metrology. The key result relies in a combination of a weak measurement with strong post-selection and the use of mixed states to increase the fluctuations of the system probe. I find this manuscript interesting, and in my opinion the result -if correct- deserves publication. However, I have found this manuscript quite technical and difficult to follow even for an audience relatively close to the quantum metrology community. First, there are several imprecisions that make confusing the reading. Namely - Eq.(1) is only correct if H depends linearly on g , whereas no linear dependence was introduced before. Moreover, the limit in terms of the variance is not tight for mixed states. In this case, the variance must be replaced by the quantum Fisher information. In addition, the authors do not mention that ΔH must be computed in the quantum initial state. ”*

Response: We are glad that the referee found our work interesting. We thank the referee for bringing this to our attention. We have changed the first paragraph accordingly.

Changes made: The beginning of the first paragraph now reads as ”Consider a physical process that is described by an interaction Hamiltonian gH , which depends linearly on a small parameter g that we want to estimate. The precision of this estimation is ultimately limited by the Cramér-Rao bound, which implies that [1]

$$\Delta g \geq \Delta g_{min} = \frac{1}{\sqrt{F(\rho)\nu}}, \quad (5)$$

where $F(\rho)$ is the quantum Fisher information (QFI) of the final state, ρ , and ν is the number of times H is used. For pure states the QFI is equal to ΔH^2 , where $\Delta H = \sqrt{\langle H^2 \rangle - \langle H \rangle^2}$ is the standard deviation of H with respect to the initial state. Hence, by preparing an initial pure state with a large ΔH , one may improve the precision.”

Comment 3.2: *“After Eq. (1), the authors write ”By preparing an initial state with a large Delta H, one can improve the precision”. This is confusing, and I understand that if we want to improve the precision, we need states with large Delta H. That is correct. However, the latter property by itself does not guarantee any success, as the completely mixed state maximizes Delta H but it is useless. ”*

Response: This is indeed correct that the completely mixed state maximizes ΔH but it is useless. We have restricted this observation for pure states and also replaced ”can improve” by ”may improve”.

Changes made: See comment 3.1.

Comment 3.3: *“After Eq. (1) the authors also comment that ”The interaction can involve a large number, N , of subsystems. In this case $H = \sum_{i=1}^N H_i$. This is very confusing because if the Hamiltonian includes interactions, such a decomposition is not possible. Actually, I do not see this is the analyzed case in subsequent sections.”*

Response: We consider the case in which the N sub-systems of the probe interact with the single system but do not interact between themselves. This is the case that we later specifically analyze and demonstrate experimentally, where the N photons of the probe interact with the single photon system.

Changes made: We have changed this sentence to "The interaction can involve a large number, N , of subsystems, and in case that there are no interactions between the subsystems, $H = \sum_{i=1}^N H_i$."

Comment 3.4: "The explanation of Fig 1 is also confusing. Graphically the figure is ok, but the authors does not comment why such scaling laws ($1/\sqrt{N}, 1/N$) are obtained. Their short explanation induces to believe they are a rather trivial consequence of the figures, and I do not see that. "

Response: Due to the comments of the referees we have realized that the discussion on the achieved scaling was not clear enough. We have made modifications in order to clarify this both in the main text and in the caption of the figure.

Changes made: In the main text we have added a paragraph that aims to give some intuition on our scheme, which is followed by a paragraph in which the quantum and classical FI are discussed. In the caption of Fig. 1 we have revised the discussion on the achieved scaling of each sub-figure. Specifically, we state that the scaling can be understood by considering the ratio between the uncertainty and the shift, which for a coherent state, a squeezed state, and a mixed state in our scheme, corresponds to $\propto 1/\sqrt{N}$, $\propto 1/N$ and $\propto 1/N$ respectively.

Comment 3.5: "Concerning the new reported results, a more detailed derivation of Eqs. (4) and (5) would be appreciated. "

Response: We agree with the referee.

Changes made: A more detailed derivation has been added to the supplementary (section II). Please note that a similar detailed derivation for the case of qubits is also provided in the supplementary (section IV).

Comment 3.6: "Secondly, the experimental analysis seems consistent, but authors need to make the approximation $Ng \ll 1$, which implies that the Heisenberg limit cannot be obtained in the form of $1/N_{total}$, being $N_{total} = N\nu$. This is the true limit imposed by quantum mechanics as demonstrated recently for instance in "V. Giovannetti et al. "Quantum Measurement Bounds beyond the Uncertainty Relations", *Phys. Rev. Lett.* 108, 260405 (2012)" and "M. J. W. Hall and H. M. Wiseman "Heisenberg-style bounds for arbitrary estimates of shift parameters including prior information", *New J. Phys.* 14, 033040 (2012)". There is no practical limit in terms of Eq. (1) as ΔH can be arbitrarily high keeping finite the mean number of photons (see the references above). Anyhow, the fact that the authors can realize precise measures reaching the Heisenberg scaling (instead of limit) by exploiting mixedness is interesting, and after a major revision the paper can be suitable for *Nature Communications*."

Response: We agree with the referee that there should be a distinction between "Heisenberg limit" and "Heisenberg scaling", and we have modified our manuscript accordingly, where "Heisenberg scaling" is being used with respect to the precision of our scheme (this is in agreement with many other experimental works, claiming Heisenberg scaling of $1/N$ and not of $1/N_{total}$).

Thanks to the comments of the referees we have made a major revision of the manuscript. We believe that the revised manuscript is greatly improved and is now suitable for publication in *Nature Communications*.

Changes made: We have changed the title of the manuscript from "Scalable Heisenberg-limited metrology using mixed states" to "Heisenberg-scaling measurement of the single-photon Kerr non-linearity using mixed states". In addition, we have substantially modified the abstract, and with respect to our scheme, we have replaced "Heisenberg limit" by "Heisenberg scaling" all along the manuscript.

Comment 3.7: *"Incidentally, I think the first reference to suggest that mixed states can outperform coherent states in the case of nonlinear optical couplings is "A. Rivas and A. Luis, "Precision Quantum Metrology and Nonclassicality in Linear and Nonlinear Detection Schemes" Phys. Rev. Lett. 105, 010403 (2010)". "*

Response: We thank the referee for bringing this to our attention.

Changes made: We have added a citation to this reference in the introduction, which is ref. [14] in the revised manuscript.

Reply to referee 4

Comment 4.1: *“The author presented and experimentally implemented a measurement scheme that utilizes the weak value. The major claim is that their scheme achieves the Heisenberg limit with only mixed initial states. I cannot recommend this manuscript to publish in Nature Communications, because I think the scheme presented by the authors does not achieve the Heisenberg-limited metrology. The reasons are given as follows: 1. In the Abstract, the authors mentioned the attempts of using $N00N$ states and squeezed states to achieve the Heisenberg limit, and then said they presented and implemented a scheme enables the Heisenberg limit to be reached. However, the “Heisenberg limit” used in the manuscript is not as the same as that in the works of using $N00N$ states and squeezed states. The authors considered a very restrictive model where the parameter to be estimated is equivalent to the strength of the coupling between a system and a probe, a post-selection is performed on the system, and the same coupling operator is measured on the probe. Furthermore, the coupling is assumed to be weak such that the measurement on the probe after the post-selection gives the imaginary part of the weak value of the coupling operator of the system. Based on this very restrictive model, the author derives a scaling behavior $1/N$ of the estimation error regarding some characteristic quantity N and named this scaling as the Heisenberg limit. However, this is different with the Heisenberg limit in the common sense, where the sensing operation is linear and independent and all kinds of measurements are considered. This can be convinced by looking at the most of the references of the manuscript. Although the scheme given in the manuscript is interesting, claiming that it achieves the Heisenberg limit, I think, is confusing and inappropriate.”*

Response: We are glad that the referee, like all of the referees, found our scheme interesting. We agree with the referee that there should be a distinction between “Heisenberg limit” and “Heisenberg scaling”, and we have modified our manuscript accordingly. Indeed, this distinction was raised by other referees as well, acknowledging the achieved, yet counter-intuitive, Heisenberg scaling of our scheme. In addition, we have emphasized the particular model of our scheme where the coupling between a pure quantum system and a probe is measured. In the experiment we have measured the Kerr non-linearity of a single photon. The quantum Fisher information (QFI) of the joint system-probe state is $\propto N^2$, where N is the average photon number of the probe. Interestingly, while in the usual method (measurement of a coherent state probe with no post-selection) the precision is SQL limited, our scheme enables to extract Fisher information (FI) that is also $\propto N^2$ by a measurement of the probe. Hence a precision with an Heisenberg scaling is achieved. The achieved precision in our scheme is lower only by a constant factor ($\simeq 5$) compared to the optimal bound (due to a pure state) set by the QFI. Moreover, in principle, using a probe state with a maximum variance ($\sim N^2$) we can attain this optimal bound.

We would like to emphasize that measuring the interaction of single photons is of special significance. Indeed, several important works have been devoted to this task, for example, Nature Photonics **3**, 95 (2008), Nature Physics **11**, 905-909 (2015), Nature Physics **13**, 540-544 (2017). However, while the precision in these works is SQL limited, we reach an Heisenberg scaling with our new technique. It is important to note that we do not use high order nonlinear self-phase modulation effect (like, for example, in Nature, 471, 486 (2011)); we consider only cross-phase modulation where the involved system is a single photon.

Changes made: We have changed the title of the manuscript from “Scalable Heisenberg-limited metrology using mixed states” to “Heisenberg-scaling measurement of the single-

photon Kerr non-linearity using mixed states”. In addition, we have substantially modified the abstract, and with respect to our scheme, we have replaced ”Heisenberg limit” by ”Heisenberg-scaling” all along the manuscript. We have significantly revised the manuscript, as well as the supplementary, where we discuss and clarify the obtained precision in terms of the QFI and the (classical) FI of our scheme.

Comment 4.2: *“2. In this work, only the post-selected events are taken into account. This is another reason that makes the claim of achieving Heisenberg limit questionable. The application of post-selection and weak value for metrology has its own significance, however, when claiming a work based on post-selection achieving the Heisenberg-limit, I believe more rigor and systematic analysis on the resource counting is necessary to make the claim convincing to the quantum metrology community.”*

Response: Due to the comments of the referees we have realized that it was not clear enough how the Heisenberg scaling is achieved. We would like to stress that the precision is always bounded by the QFI of the *joint* system-probe state. Let’s consider first a pure state of the probe (always assuming a pure state of the system), the QFI of the joint system-probe state after the interaction is $\propto N^2$, rather than $\propto N$, and this is the crucial observation here. However, if the initial state of the probe is a coherent state, then the obtained precision by measuring only the probe is limited by the SQL. For mixed probe states, the QFI is bounded by the weighted sum of the QFI of the pure joint states, which is also $\propto N^2$. Interestingly, our scheme results in a (classical) FI that is $\propto N^2$ and which is obtained by measuring the probe. Because in the usual method (measurement of a coherent state probe with no post-selection) the precision is SQL limited, our scheme is counter-intuitive, however, it does follow the limit set by the QFI. Please note that the FI of $\propto N^2$ is independent of the probability of post-selection, due to the fact that it is cancelled by the square of the weak value (Eq. S13). This is discussed in the main text (second paragraph after Eq. (5)) and in the revised supplementary (section III. B).

Changes made: In the main text we have added a paragraph on the quantum and classical Fisher information, and revised the Fisher information sections in the supplementary (sections I and II). In the revised manuscript (and supplementary) it is shown that the quantum Fisher information of the joint system-probe state is $\propto N^2$, where N is the average photon number of the probe. While a regular phase measurement of the probe yields an SQL limited precision, our scheme enables to achieve a precision with an Heisenberg scaling, namely, in our scheme, the Fisher information extracted by a measurement of the probe is $\propto N^2$. A more detailed derivation of equations (4) and (5) is given in the supplementary (section II). In addition, we have significantly revised the abstract and the summary. The independence of the FI with respect to the post-selection probability is further discussed in the revised supplementary (section III. B).

Comment 4.3: *“I also have the following technical points. 3. To derive the lower bound Eq. (1), the parameter g should be moved out of H . That is, Eq. (1) is valid for a unitary sensing $U = \exp(-igH)$ but not $U = \exp(-iH(g))$. ”*

Response: We thank the referee for this comment. We have modified this accordingly.

Changes made: The beginning of the first paragraph now reads as ”Consider a physical process that is described by an interaction Hamiltonian gH , which depends linearly on a

small parameter g that we want to estimate. The precision of this estimation is ultimately limited by the Cramér-Rao bound, which implies that [1]

$$\Delta g \geq \Delta g_{min} = \frac{1}{\sqrt{F(\rho)\nu}}, \quad (6)$$

where $F(\rho)$ is the quantum Fisher information (QFI) of the final state, ρ , and ν is the number of times H is used.”

Comment 4.4: “4. Below Eq. (2), the authors said, ”calculating the signal to noise ratio, ..., yield a precision $\Delta g \sim \Delta P^{-1}$ ”. I cannot say how this precision can be connected to the signal to noise ratio. Actually, as the author presented in the Supplementary Material, the precision, or estimation error Δg is defined through error propagation but not the signal to noise ratio. ”

Response: We agree with the referee.

Changes made: The connection between the precision and the SNR has been removed.

Comment 4.5: “5. It seems that the author used the variance of the Hamiltonian as the quantum Fisher information up to a constant multiplicative factor, see Eq. (9) of the supplementary information. However, this relation is only valid for pure states. Since the authors consider mixed probe states or classical fluctuations, I cannot see how the quantum Fisher information can be obtained from the variance of the Hamiltonian. The author should clarify the meaning of Eq. (9) and its relation to quantum Fisher information”

Response: It is indeed true that Eq. (9) of the supplementary is only valid for pure state. However, due to the convexity of the Fisher information it leads to an upper bound for the quantum Fisher information of a mixed state, which is given by the weighted sum of the quantum Fisher information of each pure state. In the revised manuscript we show that this bound on the quantum Fisher information is $\propto N^2$, and then show that the classical Fisher information extracted in our scheme is also $\propto N^2$.

Changes made: We have clarified this point along the revised manuscript (abstract, main text, and summary, see comments 4.1 and 4.2), and we have revised the supplementary which now includes a detailed derivation of the quantum and classical Fisher information.

Reviewers' comments:

Reviewer #1 (Remarks to the Author):

Nature Communications manuscript NCOMMS-17-08474A

"Heisenberg-scaling measurement of the single-photon Kerr non-linearity using mixed states" by Prof Li and colleagues

Dear Editor,

I have taken a look at the revised manuscript as well as the authors' reply to my comments as well as the other referees. There seems to be a consensus among all four referees that the manuscript reports an interesting result, but that there seems to be something inconsistent concerning the interpretation of the result when the authors claim to achieve Heisenberg scaling, and most importantly that the classical noise induced in their system is the component responsible for this scaling. Let me summarize my viewpoint in the next few paragraphs.

In optical metrology when one refers to the standard quantum limit (SQL), one is usually referring to the usual optical phase measurement, which is governed by an Hamiltonian with generator n , where n here is the photon number operator. A coherent state, which is considered the classical resource, returns a quantum Fisher information (QFI) that is $QFI \propto N$ (proportional to N), where N is the average photon number of the state. A state with some quantum resource such as entanglement or squeezing can reach $QFI \propto N^2$, which is an improvement over the SQL. This is known as "Heisenberg scaling". In this example a mixture of coherent states cannot go above the SQL, due to convexity of the QFI. Thus there is a clear distinction between the performance of classical and quantum resources.

In the present manuscript, the authors have a scenario in which they have a joint state, composed of a system (a single photon essentially behaving as a two level system) and a probe, consisting of many photons. The parameter is estimated via an interaction between the system and probe, with generator nC , where n is again the photon number operator and C is an operator on the system. As the authors show explicitly in the supplementary material, when the system is prepared in a superposition state of eigenstates of C , the QFI for ANY pure system+probe state gives a $QFI \propto N^2$. Thus, any pure probe state gives Heisenberg scaling, including the "classical" coherent states. This is shown directly in Ref. [27], just after Eqs. (9) and (10). If coherent states already give N^2 scaling, then of course a mixture of them could also give N^2 scaling, there is nothing surprising

there. In fact, for pure probes the only way to have only $\propto N$ scaling is by preparing (or measuring, presumably) the system in an eigenstate of C . Thus in this case the SQL or HL depends only on the system. In this regard it seems unfair to compare to the SQL, which is indeed a “limit” of classical probe resources only in the case of usual optical phase estimation (no coupling to an additional pure quantum system). There is some confusion in the manuscript over this point. For example, in the manuscript just after Eq. (3), when referring already to their scheme, the authors state “A coherent state $|\alpha\rangle$ has an uncertainty $\Delta n = \sqrt{N}$, for which this method yields the SQL. “

So if a coherent state probe already gives N^2 scaling, why bother using a mixture of them? How can this increase precision, as the authors claim? As is shown in the supplementary material, the QFI in the mixed probe case is bounded by $\propto \text{Var}(N) + N^2$, where Var is the variance. Here the variance and mean photon number refer to the entire probe state. For a pure coherent state probe, $\text{Var}(N) = N$, so we have $N + N^2$, which is Heisenberg scaling. By mixing coherent states one can increase the variance so that $\text{Var}(N) \propto N^2$, giving a bound that is $\propto 2 N^2$, which improves the upper bound to the QFI.

So it appears that mixing can indeed help in metrology, and this is demonstrated in the manuscript with a nice experiment. However, I feel that this nice result is completely overshadowed by the discussion and claims about Heisenberg scaling. There is indeed N^2 scaling, but this is true for any pure probe state, so in my opinion it does not make sense to make comparison to the SQL in the usual sense. In the first version of the manuscript, even after reading several times I was still under the impression that the classical noise was responsible for going beyond the $\propto N$ scaling of the pure coherent states. This incorrect viewpoint is still reinforced at some points in the present version of the manuscript, such as the discussion right after Eq. (2) and the sentence just after Eq. (3) that I cited above. I now understand that the noise helps, but is not responsible for the “scaling”.

I thus cannot recommend the manuscript for publication in its present form, as I feel that it still paints an incorrect picture of what is actually going on. Again the experiment and results are quite nice. I suggest that the authors remove this comparison with the SQL, and focus more on the improvement from mixing rather than the Heisenberg scaling. It is of course interesting in its own right that the joint system + probe can give N^2 scaling for this particular interaction. However, this was already known from Ref. [27] and others. Moreover, it is the quantum coherence of the system that is responsible for this, not the probe.

Reviewer #2 (Remarks to the Author):

The authors have provided a very comprehensive reply to the concerns detailed in my previous report and those of the other referees.

The authors conceded that 'scalable' was not an appropriate word to describe this experiment, and I am pleased to see it removed from the title. The situation is quite clear now from Figure 4, where the temporary scaling breaks down before $N=10^6$.

The authors correctly found a mistake in my reasoning about the scaling in the joint system-meter state. To me, this scaling is somewhat unintuitive in its own right, and I imagine that other readers may stumble here, too. I am glad that my confusion (which was likely not helped by my presentation in the first manuscript) has led to improvements. The response of the authors has clarified this issue greatly for me, and has indeed allayed my only remaining major concern. I must admit, however, that it is still very difficult to understand and separate out the various influences of the classical and quantum uncertainty in this particular experiment.

The arguments from convexity still hold water: it is true to say that if the laser could be held at a high photon number output (rather than fluctuating as shown in Figure 1d), the quantum Fisher information would be even higher than it was in this experiment. But in that case it could well be that the imaginary weak value approach (plus photon counting measurement) is no longer optimal, and extracting the Heisenberg scaling might require a more complicated approach. So the authors' summary that the classical noise 'is not crucial from a theoretical point of view but is vital from the practical method we used' seems to be a good one. I would suggest that they may wish to include something like this in the manuscript (for the benefit of the reader).

Along with the other improvements the authors have made, I now find this article greatly improved and suitable for publication in Nature Communications.

The authors may wish to consider the following changes:

1. Should there be units on g and Δg ? (e.g. vertical axis in Fig 4)
2. Although maximum squeezing was removed from caption of Fig1, it is still there implicitly as 'should not be greater than'. I still find this a bit confusing, as no explanation is given as to why it is practically difficult. If it is not an essential point to make, perhaps consider just leaving it out?
3. Some of the figures (particularly fig1) seem corrupted in the manuscript file (but OK in the separated figures appended to it).

4. typo below (S1). I_q should be proportional to c^2 not to c .

Reviewer #3 (Remarks to the Author):

The authors have addressed the points I raised, and have made substantial changes in the manuscript. In my opinion the major strength of the paper is the technological advance of obtaining Heisenberg-like scaling resolution for a wide range of estimated values. Therefore I believe the paper is suitable for publication.

Reviewer #4 (Remarks to the Author):

In this revised version, I believe the Authors have rather satisfactorily addressed the issues I had raised. I also believe that they have more or less satisfactorily addressed also the issues raised by the other referees. I think that the manuscript reaches the high standards required for a Nature Communication article, providing important results in the field of measuring the single-photon Kerr nonlinearity and thus I recommend it for publication as a regular article.

Besides, I also have the following minor suggestions and comments:

1. Below equation (1), the authors said: "For pure states, the QFI is equal to ΔH^2 ". This is not true. For pure states, the QFI is equal to $4\Delta H^2$.
2. To my understanding, the derivation of equation (2) needs some assumptions, e.g. g is much less than a unit. I suggest the authors give the necessary assumptions when introducing equation (2).
3. In the second line below equation (S1) of the Supplemental Information, it seems that $I_q = 4c\Delta n^2$ should be $I_q = 4c^2\Delta n^2$.
4. In equation (8), \hat{n} should be n .

Article reference: NCOMMS-17-08474A

Title: Heisenberg-scaling measurement of the single-photon Kerr non-linearity using mixed states

Authors: Geng Chen, Nati Aharon, Yong-Nan Sun, Zi-Huai Zhang, Wen-Hao Zhang, De-Yong He, Jian-Shun Tang, Xiao-Ye Xu, Yaron Kedem, Chuan-Feng Li, and Guang-Can Guo.

Reply to referee 1

Comment 1.1: *“I have taken a look at the revised manuscript as well as the authors’ reply to my comments as well as the other referees. There seems to be a consensus among all four referees that the manuscript reports an interesting result, but that there seems to be something inconsistent concerning the interpretation of the result when the authors claim to achieve Heisenberg scaling, and most importantly that the classical noise induced in their system is the component responsible for this scaling. Let me summarize my viewpoint in the next few paragraphs.”*

Response: We are very happy that all four referees found our work interesting and that three of the referees already agree that the revised manuscript is suitable for publication in Nature Communications. Referee 1 raises a concern regarding the interpretation given to the role of the classical noise in achieving the Heisenberg scaling (HS). We hope that we address this issue in a satisfactory manner in the revised manuscript.

Changes made: Please see all comments below and highlighted version of the revised manuscript.

Comment 1.2: *“In optical metrology when one refers to the standard quantum limit (SQL), one is usually referring to the usual optical phase measurement, which is governed by an Hamiltonian with generator n , where n here is the photon number operator. A coherent state, which is considered the classical resource, returns a quantum Fisher information (QFI) that is $QFI \propto N$ (proportional to N), where N is the average photon number of the state. A state with some quantum resource such as entanglement or squeezing can reach $QFI \propto N^2$, which is an improvement over the SQL. This is known as “Heisenberg scaling”. In this example a mixture of coherent states cannot go above the SQL, due to convexity of the QFI. Thus there is a clear distinction between the performance of classical and quantum resources. In the present manuscript, the authors have a scenario in which they have a joint state, composed of a system (a single photon essentially behaving as a two level system) and a probe, consisting of many photons. The parameter is estimated via an interaction between the system and probe, with generator nC , where n is again the photon number operator and C is an operator on the system. As the authors show explicitly in the supplementary material, when the system is prepared in a superposition state of eigenstates of C , the QFI for ANY pure system+probe state gives a $QFI \propto N^2$. Thus, any pure probe state gives Heisenberg scaling, including the “classical” coherent states. This is shown directly in Ref. [27], just after Eqs. (9) and (10). If coherent states already give N^2 scaling, then of course a mixture of them could also give N^2 scaling, there is nothing surprising there. In fact, for pure probes the only way to have only $\propto N$ scaling is by preparing (or measuring, presumably) the system in an eigenstate of C . Thus in this case the SQL or HL depends only on the system. In this regard it seems unfair to compare to the SQL, which is indeed a “limit” of classical probe resources only in the case of usual optical phase estimation (no coupling to an additional pure quantum*

system). There is some confusion in the manuscript over this point. For example, in the manuscript just after Eq. (3), when referring already to their scheme, the authors state “A coherent state $|\alpha\rangle$ has an uncertainty $\Delta n = \sqrt{N}$, for which this method yields the SQL.” So if a coherent state probe already gives N^2 scaling, why bother using a mixture of them? How can this increase precision, as the authors claim? ”

Response: We agree with the referee that the occurrence of photon coupling is responsible for the $\propto N^2$ scaling of QFI and “coherent states already give N^2 scaling”. The ambiguous descriptions are modified according to the referee’s comments, to make it clear that mixed states enable extracting a high FI from an interacting scenario already containing a QFI of N^2 .

Changes made: We have modified the sentences before and after Eq. (3) that mention the SQL so to avoid a possible confusion with the usual optical phase measurement that the referee describes.

Comment 1.3: “As is shown in the supplementary material, the QFI in the mixed probe case is bounded by $\propto \text{Var}(N) + N^2$, where Var is the variance. Here the variance and mean photon number refer to the entire probe state. For a pure coherent state probe, $\text{Var}(N) = N$, so we have $N + N^2$, which is Heisenberg scaling. By mixing coherent states one can increase the variance so that $\text{Var}(N) \propto N^2$, giving a bound that is $\propto 2N^2$, which improves the upper bound to the QFI. So it appears that mixing can indeed help in metrology, and this is demonstrated in the manuscript with a nice experiment. ”

Response: We thank the referee to suggest an additional benefit from using the mixed states. This improvement indeed may lead to a further improvement of the precision when all details of the scheme and the experimental set-up are taken into account.

Changes made: We have emphasized this point in the paragraph before Discussion section in the main text and also after Eq. S3 in the supplementary information.

Comment 1.4: “However, I feel that this nice result is completely overshadowed by the discussion and claims about Heisenberg scaling. There is indeed N^2 scaling, but this is true for any pure probe state, so in my opinion it does not make sense to make comparison to the SQL in the usual sense. In the first version of the manuscript, even after reading several times I was still under the impression that the classical noise was responsible for going beyond the $\propto N$ scaling of the pure coherent states. This incorrect viewpoint is still reinforced at some points in the present version of the manuscript, such as the discussion right after Eq. (2) and the sentence just after Eq. (3) that I cited above. I now understand that the noise helps, but is not responsible for the “scaling”. ”

Response: We thank the referee for giving a comprehensive understanding of the origin of N^2 scaling QFI, that is “ANY pure system+probe state gives a QFI $\propto N^2$ ”. In our scheme, the classical noise, together with the imaginary weak value, is actually responsible for extracting a FI of $\propto N^2$ by a measurement of the probe. In this revised manuscript, we make corresponding changes following the referee’s suggestions and paint an unambiguous picture to describe our anti-intuitive results.

Changes made: We have changed the sentences that could imply that the classical noise is responsible for the QFI of the joint system-probe state after the interaction, also the sentences that could lead to think that we make a comparison to the SQL in the usual

sense of an optical phase measurement described by the referee. We have added a sentence in the conclusions on the role of the classical noise, which states that "The classical noise is not crucial from a theoretical point of view but is vital for the practical method we used".

Comment 1.5: *"I thus cannot recommend the manuscript for publication in its present form, as I feel that it still paints an incorrect picture of what is actually going on. Again the experiment and results are quite nice. I suggest that the authors remove this comparison with the SQL, and focus more on the improvement from mixing rather than the Heisenberg scaling. It is of course interesting in its own right that the joint system + probe can give N^2 scaling for this particular interaction. However, this was already known from Ref. [27] and others. Moreover, it is the quantum coherence of the system that is responsible for this, not the probe."*

Response: We thank the referee to regard our experiment and results are quite nice and offer substantial helpful suggestions. Along the lines you suggest, we have removed the comparison with the SQL (only few words of SQL are reserved for the readability of the paper), and focused more on the improvement from mixing. We believe we have made it explicit that, in our scheme the classical noise enables extracting a FI that is $\propto N^2$, but it is not responsible for the QFI of $\propto N^2$ of the joint system-probe state. Ref. [27] implies another possible way to attain Heisenberg scaling in the measurement of Kerr non-linearity, however, we propose a specific strategy to extract this QFI and firstly achieve a practical Heisenberg scaling measurement.

Reply to referee 2

Comment 2.1: *“The authors have provided a very comprehensive reply to the concerns detailed in my previous report and those of the other referees. The authors conceded that ‘scalable’ was not an appropriate word to describe this experiment, and I am pleased to see it removed from the title. The situation is quite clear now from Figure 4, where the temporary scaling breaks down before $N = 10^6$. The authors correctly found a mistake in my reasoning about the scaling in the joint system-meter state. To me, this scaling is somewhat unintuitive in its own right, and I imagine that other readers may stumble here, too. I am glad that my confusion (which was likely not helped by the presentation in the first manuscript) has led to improvements. The response of the authors has clarified this issue greatly for me, and has indeed allayed my only remaining major concern. I must admit, however, that it is still very difficult to understand and separate out the various influences of the classical and quantum uncertainty in this particular experiment. The arguments from convexity still hold water: it is true to say that if the laser could be held at a high photon number output (rather than fluctuating as shown in Figure 1d), the quantum Fisher information would be even higher than it was in this experiment. But in that case it could well be that the imaginary weak value approach (plus photon counting measurement) is no longer optimal, and extracting the Heisenberg scaling might require a more complicated approach. So the authors’ summary that the classical noise ‘is not crucial from a theoretical point of view but is vital from the practical method we used’ seems to be a good one. I would suggest that they may wish to include something like this in the manuscript (for the benefit of the reader). Along with the other improvements the authors have made, I now find this article greatly improved and suitable for publication in Nature Communications. ”*

Response: We are delighted that the referee finds our manuscript suitable for publication in Nature Communications. We would like to thank the referee again for his comprehensive and constructive review.

Changes made: We have added the sentence “The classical noise is not crucial from a theoretical point of view but is vital for the practical method we used” to the conclusions.

Comment 2.2: *“The authors may wish to consider the following changes: 1. Should there be units on g and δg ? (e.g. vertical axis in Fig 4) 2. Although maximum squeezing was removed from caption of Fig1, it is still there implicitly as ‘should not be greater than’. I still find this a bit confusing, as no explanation is given as to why it is practically difficult. If it is not an essential point to make, perhaps consider just leaving it out? 3. Some of the figures (particularly fig1) seem corrupted in the manuscript file (but OK in the separated figures appended to it). 4. typo below (S1). I_q should be proportional to c^2 not to c . ”*

Response: 1. g and δg are given in radians. We have added ‘(rad)’ along the manuscript and in Fig. 4.

2. We have dropped it out.

3. We thank the referee for bringing this to our attention. We have verified the quality of the figures.

4. We thank the referee for bringing this to our attention. We have corrected this typo.

Reply to referee 3

Comment 3.1: *“The authors have addressed the points I raised, and have made substantial changes in the manuscript. In my opinion the major strength of the paper is the technological advance of obtaining Heisenberg-like scaling resolution for a wide range of estimated values. Therefoer I believe the paper is suitable for publication. ”*

Response: We are very glad that the referee finds our manuscript suitable for publication in Nature Communications. We would like to thank the referee again for his constructive review.

Reply to referee 4

Comment 4.1: *“In this revised version, I believe the Authors have rather satisfactorily addressed the issues I had raised. I also believe that they have more or less satisfactorily addressed also the issues raised by the other referees. I think that the manuscript reaches the high standards required for a Nature Communication article, providing important results in the field of measuring the single-photon Kerr nonlinearity and thus I recommend it for publication as a regular article. ”*

Response: We are very happy that the referee finds our manuscript suitable for publication in Nature Communications. We would like to thank the referee again for his constructive review.

Comment 4.2: *“Besides, I also have the following minor suggestions and comments: 1. Below equation (1), the authors said: “For pure states, the QFI is equal to ΔH^2 ”. This is not true. For pure states, the QFI is equal to $4\Delta H^2$. 2. To my understanding, the derivation of equation (2) needs some assumptions, e.g. g is much less than a unit. I suggest the authors give the necessary assumptions when introducing equation (2). 3. In the second line below equation (S1) of the Supplemental Information, it seems that $I_q = 4c\Delta n^2$ should be $I_q = 4c^2\Delta n^2$. 4. In equation (8), \hat{n} should be n . ”*

Response: 1. We thank the referee for bringing this to our attention. We have fixed this in the revised manuscript.

2. We thank the referee for bringing this to our attention. We have added this assumption before equation (2).

3. We thank the referee for bringing this to our attention. We have corrected this typo.

4. We thank the referee for bringing this to our attention. We have corrected this typo.

Reviewer #1 (Remarks to the Author):

Nature Communications manuscript NCOMMS-17-08474A

"Heisenberg-scaling measurement of the single-photon Kerr non-linearity using mixed states" by Prof Li and colleagues

Dear Editor,

I have taken a look at the revised manuscript as well as the authors' reply to my comments as well as the other referees. There seems to be a consensus among all four referees that the manuscript reports an interesting result, but that there seems to be something inconsistent concerning the interpretation of the result when the authors claim to achieve Heisenberg scaling, and most importantly that the classical noise induced in their system is the component responsible for this scaling. Let me summarize my viewpoint in the next few paragraphs.

In optical metrology when one refers to the standard quantum limit (SQL), one is usually referring to the usual optical phase measurement, which is governed by an Hamiltonian with generator n , where n here is the photon number operator. A coherent state, which is considered the classical resource, returns a quantum Fisher information (QFI) that is $QFI \propto N$ (proportional to N), where N is the average photon number of the state. A state with some quantum resource such as entanglement or squeezing can reach $QFI \propto N^2$, which is an improvement over the SQL. This is known as "Heisenberg scaling". In this example a mixture of coherent states cannot go above the SQL, due to convexity of the QFI. Thus there is a clear distinction between the performance of classical and quantum resources.

In the present manuscript, the authors have a scenario in which they have a joint state, composed of a system (a single photon essentially behaving as a two level system) and a probe, consisting of many photons. The parameter is estimated via an interaction between the system and probe, with generator nC , where n is again the photon number operator and C is an operator on the system. As the authors show explicitly in the supplementary material, when the system is prepared in a superposition state of eigenstates of C , the QFI for ANY pure system+probe state gives a $QFI \propto N^2$. Thus, any pure probe state gives Heisenberg scaling, including the "classical" coherent states. This is shown directly in Ref. [27], just after Eqs. (9) and (10). If coherent states already give N^2 scaling, then of course a mixture of them could also give N^2 scaling, there is nothing surprising there. In fact, for pure probes the only way to have only $\propto N$ scaling is by preparing (or measuring, presumably) the system in an eigenstate of C . Thus in this case the SQL or HL depends only on the system. In this regard it seems unfair to compare to the SQL, which is indeed a "limit" of classical probe resources only in the case of usual optical phase estimation (no coupling to an

additional pure quantum system). There is some confusion in the manuscript over this point. For example, in the manuscript just after Eq. (3), when referring already to their scheme, the authors state “A coherent state $|\alpha\rangle$ has an uncertainty $\Delta n = \sqrt{N}$, for which this method yields the SQL. “

So if a coherent state probe already gives N^2 scaling, why bother using a mixture of them? How can this increase precision, as the authors claim? As is shown in the supplementary material, the QFI in the mixed probe case is bounded by $\propto \text{Var}(N) + N^2$, where Var is the variance. Here the variance and mean photon number refer to the entire probe state. For a pure coherent state probe, $\text{Var}(N) = N$, so we have $N + N^2$, which is Heisenberg scaling. By mixing coherent states one can increase the variance so that $\text{Var}(N) \propto N^2$, giving a bound that is $\propto 2 N^2$, which improves the upper bound to the QFI.

So it appears that mixing can indeed help in metrology, and this is demonstrated in the manuscript with a nice experiment. However, I feel that this nice result is completely overshadowed by the discussion and claims about Heisenberg scaling. There is indeed N^2 scaling, but this is true for any pure probe state, so in my opinion it does not make sense to make comparison to the SQL in the usual sense. In the first version of the manuscript, even after reading several times I was still under the impression that the classical noise was responsible for going beyond the $\propto N$ scaling of the pure coherent states. This incorrect viewpoint is still reinforced at some points in the present version of the manuscript, such as the discussion right after Eq. (2) and the sentence just after Eq. (3) that I cited above. I now understand that the noise helps, but is not responsible for the “scaling”.

I thus cannot recommend the manuscript for publication in its present form, as I feel that it still paints an incorrect picture of what is actually going on. Again the experiment and results are quite nice. I suggest that the authors remove this comparison with the SQL, and focus more on the improvement from mixing rather than the Heisenberg scaling. It is of course interesting in its own right that the joint system + probe can give N^2 scaling for this particular interaction. However, this was already known from Ref. [27] and others. Moreover, it is the quantum coherence of the system that is responsible for this, not the probe.

Reply to referee 1

Comment 1.1: *“I have taken a look at the revised manuscript as well as the authors’ reply to my comments as well as the other referees. There seems to be a consensus among all four referees that the manuscript reports an interesting result, but that there seems to be something inconsistent concerning the interpretation of the result when the authors claim to achieve Heisenberg scaling, and most importantly that the classical noise induced in their system is the component responsible for this scaling. Let me summarize my viewpoint in the next few paragraphs. ”*

Response: We are very happy that all four referees found our work interesting and that three of the referees already agree that the revised manuscript is suitable for publication in Nature Communications. Referee 1 raises two main concerns regarding the comparison with SQL and the the role of the classical noise in achieving the Heisenberg scaling (HS). We hope that we address two issues in a satisfactory manner in the revised manuscript.

Changes made: Please see all comments below and highlighted version of the revised manuscript.

Comment 1.2: *“In optical metrology when one refers to the standard quantum limit (SQL), one is usually referring to the usual optical phase measurement, which is governed by an Hamiltonian with generator n , where n here is the photon number operator. A coherent state, which is considered the classical resource, returns a quantum Fisher information (QFI) that is $QFI \propto N$ (proportional to N), where N is the average photon number of the state. A state with some quantum resource such as entanglement or squeezing can reach $QFI \propto N^2$, which is an improvement over the SQL. This is known as “Heisenberg scaling”. In this example a mixture of coherent states cannot go above the SQL, due to convexity of the QFI. Thus there is a clear distinction between the performance of classical and quantum resources. In the present manuscript, the authors have a scenario in which they have a joint state, composed of a system (a single photon essentially behaving as a two level system) and a probe, consisting of many photons. The parameter is estimated via an interaction between the system and probe, with generator nC , where n is again the photon number operator and C is an operator on the system. As the authors show explicitly in the supplementary material, when the system is prepared in a superposition state of eigenstates of C , the QFI for ANY pure system+probe state gives a $QFI \propto N^2$. Thus, any pure probe state gives Heisenberg scaling, including the “classical” coherent states. This is shown directly in Ref. [27], just after Eqs. (9) and (10). If coherent states already give N^2 scaling, then of course a mixture of them could also give N^2 scaling, there is nothing surprising there. In fact, for pure probes the only way to have only $\propto N$ scaling is by preparing (or measuring, presumably) the system in an eigenstate of C . Thus in this case the SQL or HL depends only on the system. In this regard it seems unfair to compare to the SQL, which is indeed a “limit” of classical probe resources only in the case of usual optical phase estimation (no coupling to an additional pure quantum*

system). There is some confusion in the manuscript over this point. For example, in the manuscript just after Eq. (3), when referring already to their scheme, the authors state “A coherent state $|\alpha\rangle$ has an uncertainty $\Delta n = \sqrt{N}$, for which this method yields the SQL.” So if a coherent state probe already gives N^2 scaling, why bother using a mixture of them? How can this increase precision, as the authors claim? ”

Response: The analysis of the referee 1 is indeed correct and we agree with him that it is not adequate to make simple comparison with “usual optical phase estimation” which only yields SQL. We fix this question completely in this revised manuscript and delete all the presentations about SQL. Refer to the question of the role of mixed states, please see our response for comment 1.4.

Changes made: 1. We rewrite the abstract and delete the statements on the SQL, furthermore, we emphasize that we refer to “a photon-coupling scenario” thus “the QFI shows a quantum-enhanced scaling of N^2 ”, which is different from “usual optical phase estimation”.

2. We delete all the presentation relevant to SQL in the introduction.

3. We amend Fig. 1 and delete previous panel (a) and (b), which mainly describe “usual optical phase estimation” related to SQL.

4. All sentences mentioning SQL when we discuss the experiment results in the Results section are also deleted.

Comment 1.3: “ As is shown in the supplementary material, the QFI in the mixed probe case is bounded by $\propto \text{Var}(N) + N^2$, where Var is the variance. Here the variance and mean photon number refer to the entire probe state. For a pure coherent state probe, $\text{Var}(N) = N$, so we have $N + N^2$, which is Heisenberg scaling. By mixing coherent states one can increase the variance so that $\text{Var}(N) \propto N^2$, giving a bound that is $\propto 2N^2$, which improves the upper bound to the QFI. So it appears that mixing can indeed help in metrology, and this is demonstrated in the manuscript with a nice experiment. ”

Response: We thank the referee to suggest an additional benefit from using the mixed states. This improvement indeed may lead to a further improvement of the precision when all details of the scheme and the experimental set-up are taken into account.

Changes made: We have emphasized this point in (I) the abstract by adding a new sentence that “Benefit from the use of mixed state, the upper bound of QFI is improved to $2N^2$ ”; (II) the second paragraph of introduction; (III) the paragraph before the Discussion section; (IV) the supplementary information after Eq. S3.

Comment 1.4: “However, I feel that this nice result is completely overshadowed by the discussion and claims about Heisenberg scaling. There is indeed N^2 scaling, but this is true for any pure probe state, so in my opinion it does not make sense to make comparison to the SQL in the usual sense. In the first version of the manuscript, even after reading several times I was still under the impression that the classical noise was responsible for going beyond the $\propto N$ scaling of the pure coherent states. This incorrect viewpoint is still reinforced at some points in the present version of the manuscript, such as the discussion right after Eq. (2) and the sentence just after Eq. (3) that I cited above. I now understand that the noise helps, but is not responsible for the “scaling”. ”

Response: We thank the referee to think our result is nice. We agree with his comprehensive understanding of the origin of N^2 scaling QFI, that is "ANY pure system+probe state gives a QFI $\propto N^2$ ". However, one should note that the QFI only gives a theoretical upper bound on the measurement precision, in other words, a N^2 scaling of QFI will not give a practical Heisenberg scaling automatically. One has to devise an experimentally feasible scheme that would yield a high FI. In our scheme, the classical noise, together with the imaginary weak value, are actually responsible for attaining a FI of $\propto N^2$ by a measurement of the probe.

Although ref. [27] reveals it is theoretically possible to reach Heisenberg-scaling in the measurement of Kerr non-linearity, as far as we know, no experimental work has reached such a practical Heisenberg-scaling. In this work, just by utilizing mixed state, we propose a specific strategy to extract a FI of $\sim N^2$ and for the first time achieve a practical Heisenberg scaling measurement. The referee's confusion on this point maybe partially due to the ambiguous description in previous manuscript, so we make considerable changes on this issue.

Changes made: We have changed the sentences that could imply that the classical noise is responsible for the QFI of the joint system-probe state after the interaction. The changes includes:

1. In the abstract, we emphasize the main significance of our work is that we design a specific strategy to extract a QFI of $\sim N^2$.
2. In the introduction, we reinforce the point the mixed state together with a postselection is necessary for obtaining a practical Heisenberg-scaling.
3. Following referee 2's suggestion, we have added a sentence in the conclusions on the role of the classical noise, which states that "The classical noise is not crucial from a theoretical point of view but is vital for the practical method we used"

Comment 1.5: *"I thus cannot recommend the manuscript for publication in it's present form, as I feel that it still paints an incorrect picture of what is actually going on. Again the experiment and results are quite nice. I suggest that the authors remove this comparison with the SQL, and focus more on the improvement from mixing rather than the Heisenberg scaling. It is of course interesting in it's own right that the joint system + probe can give N^2 scaling for this particular interaction. However, this was already known from Ref. [27] and others. Moreover, it is the quantum coherence of the system that is responsible for this, not the probe."*

Response: We thank the referee to regard our experiment and results are quite nice and offer substantial helpful suggestions. Along the lines he suggests, we have removed all the comparison with the SQL, and focused more on the improvement from mixing. Meanwhile, we make it more explicit that, in our scheme the classical noise enables attaining an FI that is $\propto N^2$, but it is not responsible for the QFI of $\propto N^2$ of the joint system-probe state. Rather than a theoretical inference, we propose a specific strategy to extract this QFI and for the first time achieve a practical Heisenberg scaling measurement.

Reply to referee 2

Comment 2.1: *“The authors have provided a very comprehensive reply to the concerns detailed in my previous report and those of the other referees. The authors conceded that ‘scalable’ was not an appropriate word to describe this experiment, and I am pleased to see it removed from the title. The situation is quite clear now from Figure 4, where the temporary scaling breaks down before $N = 10^6$. The authors correctly found a mistake in my reasoning about the scaling in the joint system-meter state. To me, this scaling is somewhat unintuitive in its own right, and I imagine that other readers may stumble here, too. I am glad that my confusion (which was likely not helped by the presentation in the first manuscript) has led to improvements. The response of the authors has clarified this issue greatly for me, and has indeed allayed my only remaining major concern. I must admit, however, that it is still very difficult to understand and separate out the various influences of the classical and quantum uncertainty in this particular experiment. The arguments from convexity still hold water: it is true to say that if the laser could be held at a high photon number output (rather than fluctuating as shown in Figure 1d), the quantum Fisher information would be even higher than it was in this experiment. But in that case it could well be that the imaginary weak value approach (plus photon counting measurement) is no longer optimal, and extracting the Heisenberg scaling might require a more complicated approach. So the authors’ summary that the classical noise ‘is not crucial from a theoretical point of view but is vital from the practical method we used’ seems to be a good one. I would suggest that they may wish to include something like this in the manuscript (for the benefit of the reader). Along with the other improvements the authors have made, I now find this article greatly improved and suitable for publication in Nature Communications. ”*

Response: We are delighted that the referee finds our manuscript suitable for publication in Nature Communications. We would like to thank the referee again for his comprehensive and constructive review.

Changes made: We have added the sentence “The classical noise is not crucial from a theoretical point of view but is vital for the practical method we used” to the conclusions.

Comment 2.2: *“The authors may wish to consider the following changes: 1. Should there be units on g and δg ? (e.g. vertical axis in Fig 4) 2. Although maximum squeezing was removed from caption of Fig1, it is still there implicitly as ‘should not be greater than’. I still find this a bit confusing, as no explanation is given as to why it is practically difficult. If it is not an essential point to make, perhaps consider just leaving it out? 3. Some of the figures (particularly fig1) seem corrupted in the manuscript file (but OK in the separated figures appended to it). 4. typo below (S1). I_q should be proportional to c^2 not to c . ”*

Response: 1. g and δg are given in radians. We have added ‘(rad)’ along the manuscript and in Fig. 4.

2. We have dropped it out.

3. We thank the referee for bringing this to our attention. We have verified the quality of the figures.

4. We thank the referee for bringing this to our attention. We have corrected this typo.

Reply to referee 3

Comment 3.1: *“The authors have addressed the points I raised, and have made substantial changes in the manuscript. In my opinion the major strength of the paper is the technological advance of obtaining Heisenberg-like scaling resolution for a wide range of estimated values. Therefore I believe the paper is suitable for publication.”*

Response: We are very glad that the referee finds our manuscript suitable for publication in Nature Communications. We would like to thank the referee again for his constructive review.

Reply to referee 4

Comment 4.1: *“In this revised version, I believe the Authors have rather satisfactorily addressed the issues I had raised. I also believe that they have more or less satisfactorily addressed also the issues raised by the other referees. I think that the manuscript reaches the high standards required for a Nature Communication article, providing important results in the field of measuring the single-photon Kerr nonlinearity and thus I recommend it for publication as a regular article.”*

Response: We are very happy that the referee finds our manuscript suitable for publication in Nature Communications. We would like to thank the referee again for his constructive review.

Comment 4.2: *“Besides, I also have the following minor suggestions and comments: 1. Below equation (1), the authors said: “For pure states, the QFI is equal to ΔH^2 ”. This is not true. For pure states, the QFI is equal to $4\Delta H^2$. 2. To my understanding, the derivation of equation (2) needs some assumptions, e.g. g is much less than a unit. I suggest the authors give the necessary assumptions when introducing equation (2). 3. In the second line below equation (S1) of the Supplemental Information, it seems that $I_q = 4c\Delta n^2$ should be $I_q = 4c^2\Delta n^2$. 4. In equation (8), \hat{n} should be n .”*

Response: 1. We thank the referee for bringing this to our attention. We have fixed this in the revised manuscript.

2. We thank the referee for bringing this to our attention. We have added this assumption before equation (2).

3. We thank the referee for bringing this to our attention. We have corrected this typo.

4. We thank the referee for bringing this to our attention. We have corrected this typo.

REVIEWERS' COMMENTS:

Reviewer #1 (Remarks to the Author):

Dear Editor,

The authors have adequately addressed my concerns about the presentation of their results. In my opinion, the manuscript could be published in Nature Communications.

Reply To Referee 1

Referee 1: The authors have adequately addressed my concerns about the presentation of their results. In my opinion, the manuscript could be published in Nature Communications.

Our Response: We are glad that referee 1 satisfies with our modifications and we thank his recommendation for publication in Nature Communications.